# Drp1 splice variants regulate ovarian cancer mitochondrial dynamics and tumor progression

Zaineb Javed[1,2,3], Dong Hui Shin[3,14], Weihua Pan[1,2], Sierra R White[2,4], Amal Taher Elhaw[1,2,3], Yeon Soo Kim[3,15], Shriya Kamlapurkar [1,2], Ya-Yun Cheng[1,2], J Cory Benson [5], Ahmed Emam Abdelnaby [5], Rébécca Phaëton[6,16], Hong-Gang Wang[7], Shengyu Yang[8], Mara L G Sullivan [9], Claudette M St.Croix [9], Simon C Watkins[9], Steven J Mullett[5,10], Stacy L Gelhaus [5,10], Nam Lee [11], Lan G Coffman[1,2], Katherine M Aird[1,3], Mohamed Trebak [1,3,4], Karthikeyan Mythreye [12], Vonn Walter [13] & Nadine Hempel [1,2,4✉]

## Abstract

**Aberrant mitochondrial fission/fusion dynamics are frequently associated with pathologies, including cancer. We show that alternative splice variants of the fission protein Drp1 (*DNM1L*) contribute to the complexity of mitochondrial fission/fusion regulation in tumor cells. High tumor expression of the Drp1 alternative splice variant lacking exon 16 relative to other transcripts is associated with poor outcome in ovarian cancer patients. Lack of exon 16 results in Drp1 localization to microtubules and decreased association with mitochondrial fission sites, culminating in fused mitochondrial networks, enhanced respiration, changes in metabolism, and enhanced pro-tumorigenic phenotypes in vitro and in vivo. These effects are inhibited by siRNAs designed to specifically target the endogenously expressed transcript lacking exon 16. Moreover, lack of exon 16 abrogates mitochondrial fission in response to pro-apoptotic stimuli and leads to decreased sensitivity to chemotherapeutics. These data emphasize the pathophysiological importance of Drp1 alternative splicing, highlight the divergent functions and consequences of changing the relative expression of Drp1 splice variants in tumor cells, and strongly warrant consideration of alternative splicing in future studies focused on Drp1.**

**Keywords** Drp1; *DNM1L*; Mitochondrial Fission; Ovarian Cancer; Alternative Splice Variants
**Subject Categories** Cancer; Organelles

## Introduction

Mitochondria are highly dynamic organelles continuously undergoing fission and fusion events to facilitate adaptations to cellular and extracellular cues. The opposing processes of mitochondrial fission and fusion are mediated by several evolutionarily conserved dynamin-related GTPases, including Mitofusins (MFN1&2) and OPA1, which promote fusion, and Dynamin-related Protein 1 (Drp1, encoded by *DNM1L*) which mediates mitochondrial fission (Archer, 2013; Kamerkar et al, 2018). Drp1 binds to mitochondrial anchor proteins and forms homo-multimeric helical structures around the outer mitochondrial membrane to initiate division. This process is tightly regulated by Drp1 post-translational modifications (Cribbs and Strack, 2007; Santel and Frank, 2008; Strack and Cribbs, 2012; Yamada et al, 2016; Yu et al, 2021). Given that the shape of mitochondria are inextricably linked to their function, maintaining a balance between these dynamic fission and fusion events is essential in preserving mitochondrial respiration, and for the proper distribution of mitochondria and mitochondrial DNA during mitosis (Benard et al, 2007; Parone et al, 2008). Moreover, mitochondrial fission is also an integral component of the apoptotic and mito/autophagy pathways (Arnoult et al, 2005; Twig et al, 2008). Perturbations in mitochondrial fission/fusion dynamics have been implicated in cancer, and cancer cells can exploit adaptive mitochondrial dynamics to their advantage to meet heightened energy demands and to regulate cellular processes, including tumor metabolism, stress response pathways and resistance to apoptosis. Several studies have described a role for enhanced fission in cancer (Kashatus et al, 2015; Nagdas et al, 2019; Serasinghe et al, 2015; Tanwar et al, 2016), and this shown to be associated with cell cycle progression (Rehman et al, 2012; Taguchi et al, 2007) and migration

[1]UPMC Hillman Cancer Center, University of Pittsburgh School of Medicine, Pittsburgh, PA, USA. [2]Department of Medicine, Division of Hematology/Oncology, University of Pittsburgh School of Medicine, Pittsburgh, PA, USA. [3]Department of Pharmacology, College of Medicine, Pennsylvania State University, Hershey, PA, USA. [4]Vascular Medicine Institute (VMI), University of Pittsburgh School of Medicine, Pittsburgh, PA, USA. [5]Department of Pharmacology and Chemical Biology, University of Pittsburgh School of Medicine, Pittsburgh, PA, USA. [6]Department of Obstetrics & Gynecology, College of Medicine, Pennsylvania State University, Hershey, PA, USA. [7]Department of Pediatrics, College of Medicine, Pennsylvania State University, Hershey, PA, USA. [8]Department of Cellular and Molecular Physiology, College of Medicine, Pennsylvania State University, Hershey, PA, USA. [9]Center for Biologic Imaging, University of Pittsburgh School of Medicine, Pittsburgh, PA, USA. [10]Health Sciences Mass Spectrometry Core, University of Pittsburgh, Pittsburgh, PA, USA. [11]Division of Pharmacology, Chemistry and Biochemistry, College of Medicine, University of Arizona, Tucson, AZ, USA. [12]Department of Pathology and O'Neal Comprehensive Cancer Center, Heersink School of Medicine, University of Alabama at Birmingham, Birmingham, AL, USA. [13]Department of Public Health Sciences, Division of Biostatistics and Bioinformatics and Department of Biochemistry and Molecular Biology, College of Medicine, Pennsylvania State University, Hershey, PA, USA. [14]Present address: School of Pharmacy, Virginia Commonwealth University, Richmond, VA, USA. [15]Present address: Division of Human Biology, Fred Hutchinson Cancer Center, Seattle, WA, USA. [16]Present address: GlaxoSmithKline, Collegeville, PA, USA. ✉E-mail: nah158@pitt.edu

(Desai et al, 2013). Underlying genetic factors may also contribute to exacerbated signaling that drives the activation of fission, as observed in BRAF and KRAS mutant tumors where Drp1 S616phosphorylation is activated via Erk (Kashatus et al, 2015; Nagdas et al, 2019; Serasinghe et al, 2015; Wieder et al, 2015). On the contrary, mitochondrial fission is associated with enhanced apoptosis and decreased fission can therefore result in apoptosis resistance (Arnoult et al, 2005; Caino et al, 2015; Frank et al, 2001; Thomas and Jacobson, 2012). Mitochondrial fusion may thus be one mechanism by which tumors evade apoptosis in response to chemotherapeutic agents (Farrand et al, 2013; Kong et al, 2014b). These seemingly conflicting studies appear to suggest that both aberrant fission and fusion could have important roles during tumor progression and that the dysregulation of mitochondrial fission and fusion dynamics has the potential to differentially influence various disease stages, including progression, recurrence, and chemoresistance.

In addition to regulation by post-translational mechanisms, alternative splicing of the *DNM1L* pre-mRNA transcript has been shown to give rise to Drp1 splice variants with differential tissue expression, subcellular localization, and fission activity (Chen et al, 2000; Howng et al, 2004; Itoh et al, 2018; Macdonald et al, 2016; Rosdah et al, 2020; Strack et al, 2013). Despite the established importance of Drp1 as an integral mitochondrial fission protein, few studies have investigated the expression and unique functions of Drp1 splice variants in pathophysiological contexts. While Drp1 has been shown to be overexpressed in several tumor types, the role of individual splice variants was not considered in these studies. In ovarian cancer *DNM1L* gene amplification has been associated with poor patient outcomes (Tanwar et al, 2016; Tsuyoshi et al, 2020). However, it is unclear how this reflects the relative expression and function of Drp1 splice variants. Here, we show for the first time that transcripts arising from exon 16 splicing are highly expressed in ovarian cancer cells, and that relative expression of this transcript to full-length Drp1 mRNA is predictive of poor patient outcome. We demonstrate that exon 16 splicing results in a Drp1 protein with a unique function related to the regulation of mitochondrial architecture, mitochondrial function, tumor metabolism, and chemosensitivity, which is coopted by aggressive ovarian cancer cells, and that relative expression of different Drp1 splice variants has consequences on mitochondrial function and tumor progression. This represents the first study demonstrating the pathophysiological relevance of Drp1 splice variants.

## Results

### Ovarian cancer cells display distinct Drp1/*DNM1L* splice variant expression

We previously observed that ovarian cancer cell lines express several different molecular weight protein variants of Drp1 (Dier et al, 2014). To determine the clinical significance of this observation, Drp1 protein expression was assessed in ascites-derived epithelial ovarian cancer (EOC) cells isolated from ovarian cancer patients (Fig. 1A). Four bands around the molecular weight of 75 kDa were detected using a polyclonal antibody raised against the variable and GED domain of Drp1 (Millipore ABT155). Of these, two protein bands were prominently expressed (indicated by green and red arrows), with propensity for the lower molecular weight band (red arrow) being present in samples of high-grade serous ovarian adenocarcinoma origin (HGSA: ECO7, EOC14, and EOC15; Fig. 1A). Two major Drp1

protein bands were also observed in OVCA420 and OVCA433 ovarian cancer cells lines and these validated as being Drp1 using siRNA-mediated knock-down (Fig. 1B).

Several Drp1/*DNM1L* transcript variants are annotated on the RefSeq record, including alternative splicing of exons 3, 16, and 17 (Howng et al, 2004; Strack and Cribbs, 2012; Strack et al, 2013; Uo et al, 2009; Yoon et al, 1998). To determine if the observed Drp1 protein variants are due to alternative start site utilization, splicing, or alternate transcriptional termination, 5′ and 3′ rapid amplification of cDNA ends (RACE) was carried out. 3′RACE revealed that ovarian cancer cells express multiple Drp1 transcripts, including mRNAs with alternatively spliced exons 16 and 17, differential 3′UTR lengths, and short transcripts that terminate after exon 14 (ΔC-Ex14) and in intron 17 (ΔC-In17) at predicted alternate polyadenylation sites (Fig. 1C,D; Appendix Fig. S1A,C). 5′RACE demonstrated utilization of the same transcriptional start site and that all transcripts do not contain exon 3 (Appendix Fig. S1B,C). The C-terminal truncated transcript ΔC-In17 displayed alternative splicing of exon 16 and had a predicted alternate stop codon within intron 17, leading to a transcript with a novel coding sequence for 16 amino acids derived from intron 17 (Appendix Fig. S1C,D). This was predicted to express a 65 kDa protein lacking the C-terminal GED domain (Appendix Fig. S1D). Using a polyclonal antibody, we were able to detect a protein at the predicted size, which was decreased in expression following siRNA mediate Drp1 knock-down (Appendix Fig. S1E), suggesting that the ΔC-In17 transcript may result in expression of a truncated protein. ΔC-In17 was detected to variable degrees in other ovarian cancer cell lines and patient-derived tumor cells by RT-PCR (Appendix Figs. S1F, S2A,B). Annotation of TCGA ovarian cancer data for the ΔC-In17 transcripts demonstrated that these were detected in <15% of TCGA ovarian cancer specimens (Fig. 2A). While the functional consequences of this novel yet rare C-terminal truncation transcript require further study, we focused on further elucidating the function of alternative splice variants of the variable domain exons 16 and 17 in full-length Drp1, as their predicted proteins molecular weights matched the predominant protein variants observed in patient ascites-derived EOCs and ovarian cancer cell lines (Fig. 1A,B).

Exons 16 and 17 are located in the variable B-domain of Drp1 (Fig. 1E), and alternative splicing of these exons (denoted as 16/17) was further examined using RT-PCR with primers flanking this region. Variable expression of splice variants was found in a panel of ovarian cancer cell lines, with the HGSA lines OVCAR3, OVCA420 and OVCA433 demonstrating higher relative expression of the transcript with exon 16 spliced-out, referred hereafter as Drp1(-/17) (Fig. 1F). From here we denote the variable domain region of Drp1 as (16/17) with exons spliced out marked by a dash (-) (Fig. 1E).

### Patient ascites-derived epithelial ovarian cancer cells and tumor specimens display high expression of the Drp1 transcript variant lacking exon 16, which is associated with poor patient outcome

By annotating TCGA RNA sequencing data for the identified Drp1/ *DNM1L* transcripts, we found that all variable domain (Exons 16 and 17) splice variants of the full-length transcripts were expressed in Ovarian Serous Cystadenocarcinoma TCGA specimens, albeit at different levels (Fig. 2A,B). Of the four variable domain variants, the highest expression of Drp1(-/17), the transcript lacking exon 16, was observed, followed by Drp1(16/17) and Drp1(-/-) displaying approximately equal expression (Fig. 2B). Drp1(16/-), the transcript lacking exon 17, was least abundant. Similar to our

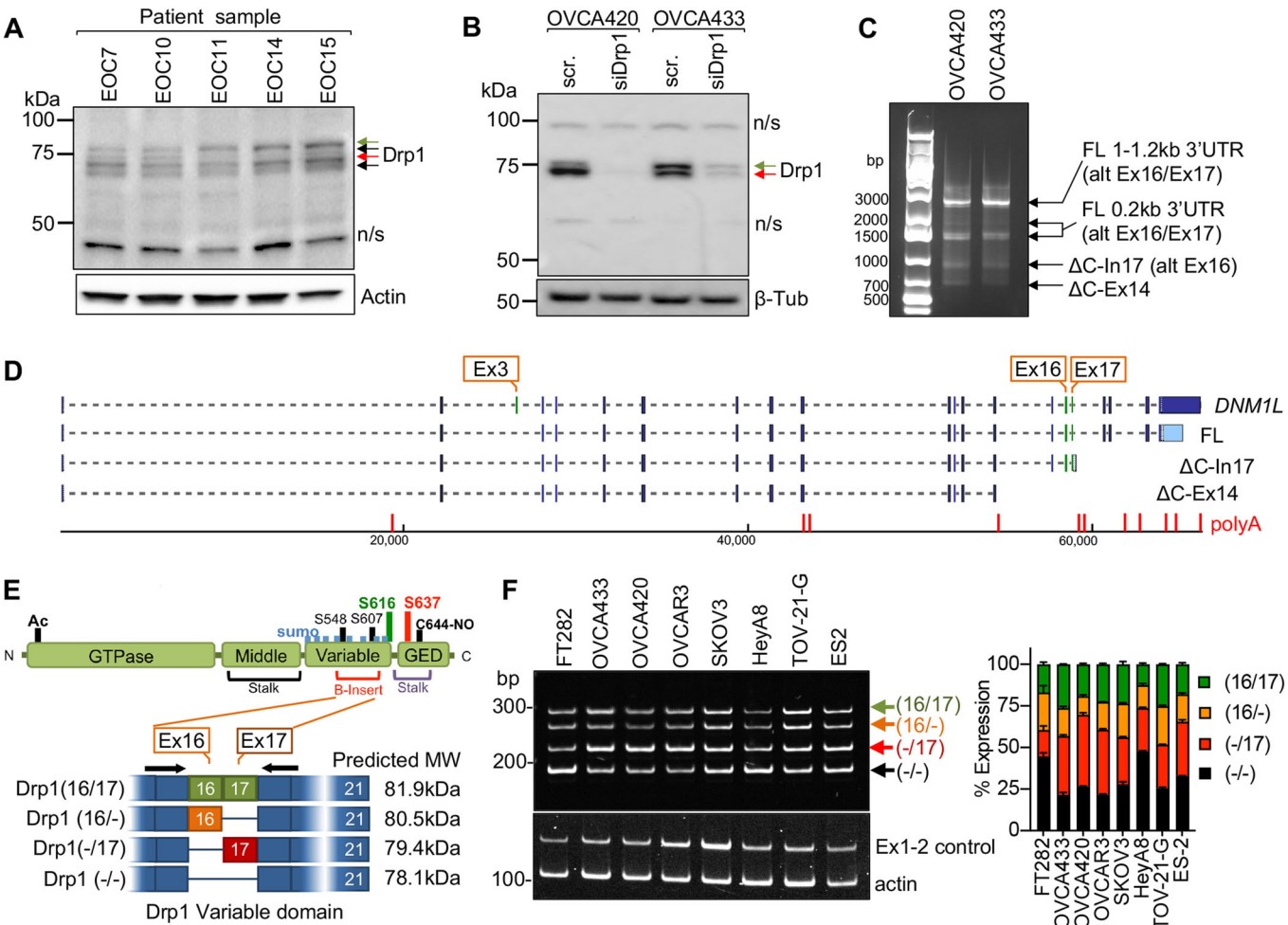

**Figure 1. Ovarian cancer cells express splice variants of Drp1/*DNM1L*.**

(A) Western blot analysis of Drp1 protein expression in patient ascites-derived epithelial ovarian cancer cells (EOC), with the following histological classification: EOC7: HGSA stage IC; EOC10: granulosa tumor IV; EOC 11: carcinosarcoma stage IIIB; EOC14: HGSA stage IV; EOC15: HGSA stage IV. Arrows point to the predicted molecular weight protein (green arrow) and a lower molecular weight band (red arrow), that is prominently expressed in EOCs from patient ascites (Drp1 antibody: ABT155). (B) Drp1 protein variants (arrows) were identified in OVCA420 and OVCA433 cells by western blotting using the N-terminal anti-Drp1 monoclonal antibody ab184247. Specificity to Drp1 was assessed by siRNA-mediated knock-down. Potential non-specific bands (n/s) are indicated. One representative blot from three independent replicates is shown. (C) 3′ RACE and subsequent sequencing of PCR products reveals that OVCA420 and OVCA433 cells express multiple *DNM1L* transcripts variants, including full-length (FL) transcripts with alternatively spliced exons 16 and 17, and C-terminal truncated transcripts at exon 14 (ΔC-Ex14) and intron 17 (ΔC-In17). (D) Schematic of transcript variants identified in OVCA420 and OVCA433 cells by 5′ and 3′ RACE (from panel C), including alternative splicing of the variable domain exons 16 and 17 (panel C: alt Ex 16/17); variable lengths of 3′UTRs (panel C: FL 1–1.2 kb, 0.2 kb 3′UTR), and utilization of alternate proximal polyadenylation, resulting in two C-terminal truncation variants, terminating in Intron 17 (ΔC-In17) and exon 14 (ΔC-Ex14). ΔC-In17 has two variants due to exon 16 alternative splicing and has a predicted STOP codon following a novel coding sequence for 16 amino acids within intron 17. (E) Schematic representation of functional domains and areas of post-translational modification of the Drp1/*DNM1L* protein. The location of alternative spliced exons 16 and 17 is in the variable B-insert domain. Numbers in brackets of transcript names denote included exons of the variable domain, dash denotes exon is spliced out. (F) RT-PCR with primers flanking the variable domain illustrates the relative expression of the four *DNM1L* variable domain splice variants derived from alternative splicing of exons 16 and 17 in ovarian cancer cell lines (mean ± SEM from three independent cultures). Primers in exon 1–2 are used to detect total Drp1. Source data are available online for this figure.

findings in cell lines, most Drp1/*DNM1L* transcripts in TCGA specimens lacked exon 3 (Fig. EV1A), agreeing with previous work describing exon 3 retention to be predominant in neuronal tissues (Strack et al, 2013). TCGA data were independently validated by assessing splice variant transcript abundance in EOCs isolated from patient ascites (Fig. 2C) and in a separate cohort of matched normal fallopian and omental or ovarian tumor specimens (Fig. 2D). Ascites-derived EOCs classified as HGSA and carcinosarcoma demonstrated predominant expression of the splice variant lacking

exon 16, Drp1(-/17), and transcripts containing both exons 16 and 17, Drp1(16/17). In addition, an increase in relative Drp1(-/17) expression was observed in ovarian tumors (11/13 specimens) and omental tumors (5/5) compared to matched benign/normal fallopian tubes (Fig. 2D; Appendix Fig. S2B). These data demonstrate that ovarian cancer cells express several Drp1/*DNM1L* splice variants, with a high abundance of Drp1(-/17) expression, and suggest that splicing of exon 16 and retention of exon 17 might be of significance to ovarian cancer.

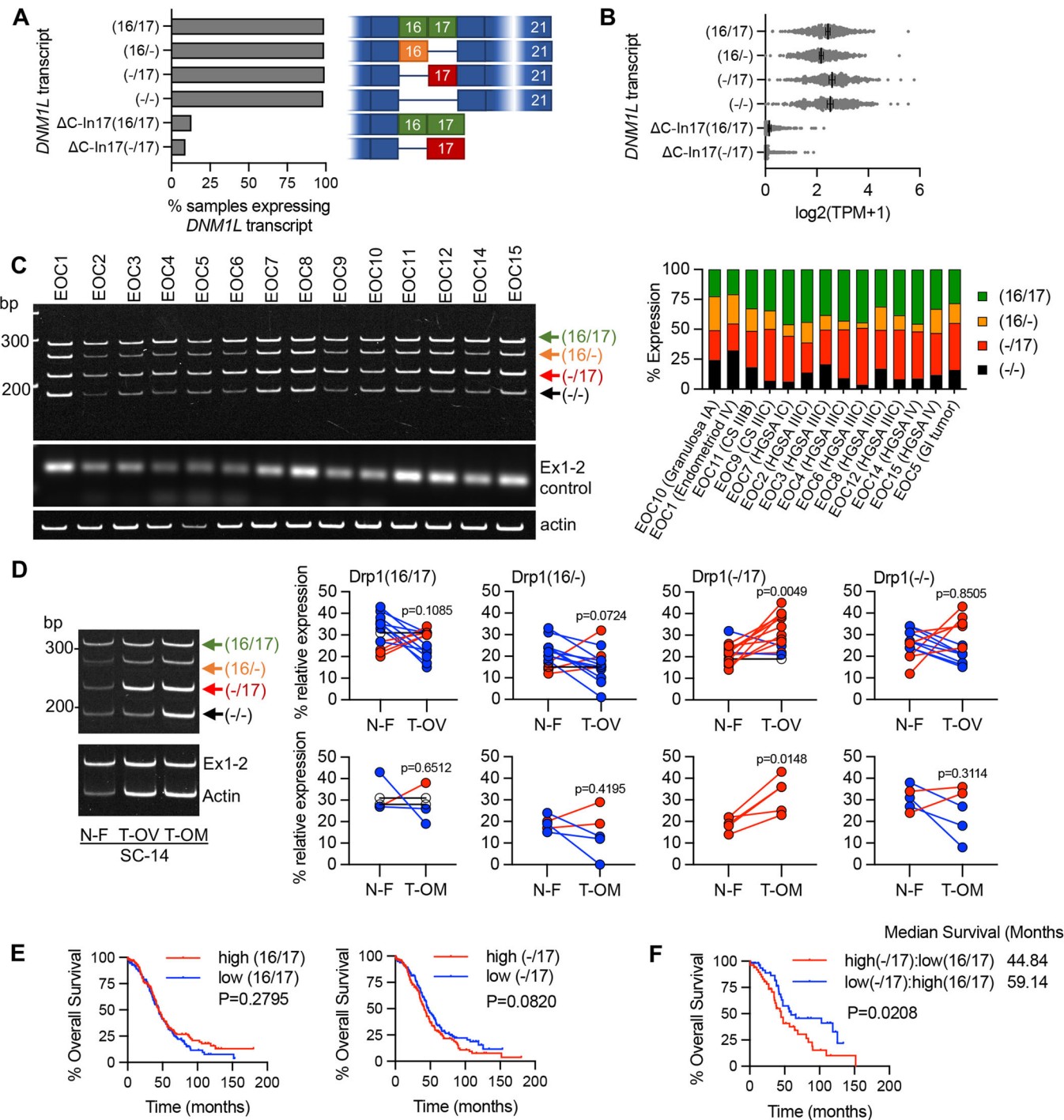

We were curious if the predominance of Drp1 splice variant expression is predictive of patient outcome. Comparisons between high and low expression of individual Drp1 splice variants was not able to significantly predict overall survival (Figs. 2E and EV1B). However, grouping patients into mutually exclusive high vs low expression of splice variant pairs based on median expression cut-offs, revealed that relative abundance of the different variable domain splice variants influenced patient outcome (Figs. 2F and EV1C,D). Patients with tumors displaying high Drp1(-/17) and low

Drp1(16/17) expression demonstrated poor overall survival compared to those with high Drp1(16/17) and low Drp1(-/17) expression (HR: 1.81, 95% CI: 1.076 to 3.071, log-rank $p = 0.0208$), with a median survival difference of 14.3 months (Fig. 2F). High Drp1(-/17):low Drp1(16/-) expression also decreased median survival by 11.7 months, albeit not significantly (HR: 1.508, 95% CI: 0.9304 to 2.444, log-rank $p = 0.088$; Fig. EV1D). Other mutually exclusive expression comparisons between Drp1 splice variants were not significantly predictive of survival

**Figure 2. Drp1/*DNM1L* transcript variant expression in ovarian cancer specimens.**

(A) Frequency of Drp1/*DNM1L* transcript variant expression in ovarian serous cystadenocarcinoma TCGA specimens, focusing on full-length variable domain (16/17) transcripts and C-terminal truncations terminating in Intron 17 (ΔC-In17). Data represent the percentage of specimens displaying TPM values >0.5 for each *DNM1L* transcript variant. (B) Expression levels of *DNM1L* transcript variants expressed as log2(TPM + 1) values from 368 individual ovarian serous cystadenocarcinoma TCGA patient samples (median with 95% CI). (C) RT-PCR was used to show the relative expression of *DNM1L* variable domain splice variants in a panel of patient ascites-derived EOCs. Histologic classification and stage indicated in graph (right; CS carcinosarcoma, HGSA high-grade serous adenocarcinoma, GI gastrointestinal). (D) Representative RT-PCR (left) of *DNM1L* variable domain splice variant expression from normal fallopian tube (N-F), and matched ovarian (T-OV) and omental tumors (T-OM). The relative expression of splice variant transcript Drp1(-/17) is consistently higher in ovarian tumor and omental tumor compared to matched normal fallopian tube specimens (blots see Appendix Fig. S2; blue lines indicate decreased expression, red lines indicate increased expression, and black lines indicate no change in expression relative to matched normal fallopian tube tissue, N-F vs T-OV $n = 13$, N-F vs T-OM $n = 5$; paired *t*-test). (E) Overall survival of TCGA ovarian cancer patients based on DNM1L variant expression. Samples were split at median log2 TPM into high ($n = 184$) and low expression ($n = 184$; log-rank Mantel-Cox test). (F) Overall survival comparison between samples displaying mutually exclusive high Drp1(-/17)/low Drp1(16/17) ($n = 52$) and low Drp1(-/17)/high Drp1(16/17) ($n = 52$) expression (low and high cutoffs based on median log2(TPM + 1); log-rank Mantel-Cox test). Source data are available online for this figure.

(Fig. EV1D). Although Drp1(16/17) and Drp1(-/17) transcript variants are both abundant in ovarian cancer cell lines and patient specimens, the above data suggest that their relative expression has consequences on ovarian cancer progression. Yet, the significance of exon 16 splicing on the Drp1 function had not previously been explored in cancer.

## Loss of Exon 16 abrogates association with mitochondrial fission puncta and leads to more fused mitochondrial networks

The HGSA OVCA433 cell lines was chosen as a model for subsequent studies as it displayed similar *DNM1L* variable domain transcript expression as HGSA patient specimens, while SKOV3 cells were chosen as a model due to their relatively equal expression of all four full-length variable domain variants. To investigate their function, Drp1 splice variants were expressed as GFP fusion proteins by lentiviral transduction (Figs. 3A and EV2A). Drp1(16/17) which contains exons 16 and 17 was localized to the cytosol and at distinct mitochondrial fission puncta (Figs. 3B and EV2B), as previously described by numerous Drp1 overexpression studies (Cribbs and Strack, 2007; Michalska et al, 2018; Strack et al, 2013). On the contrary, Drp1(-/17) displayed less mitochondrial association and fewer aggregates at fission puncta. Instead, Drp1(-/17) exhibited a filamentous pattern of localization (Figs. 3B and EV2B). Consistent with the work by Strack and Cribbs, who demonstrated that the splicing of exon 16 modifies the association of Drp1 with mitochondria and increases binding to microtubules (Strack et al, 2013), we found that Drp1(-/17) exhibited overlapping staining with tubulin in ovarian cancer cell lines (Figs. 3B,C and EV2B,C).

Expression of Drp1(-/17) was previously reported to lead to more fused and elongated mitochondrial networks, likely due to the reduced association of Drp1(-/17) with mitochondria (Strack et al, 2013). In line with this, compared to both GFP control and Drp1(16/17), cells expressing Drp1(-/17) had longer average mitochondrial branch lengths and a greater number of branches per mitochondria, indicative of a more interconnected mitochondrial network (Figs. 3D and EV2D). In contrast, expression of Drp1(16/17) led to shorter branch lengths and fewer branches per mitochondria, a predicted phenotype following overexpression of Drp1 (Figs. 3D and EV2D). TEM imaging additionally demonstrated elongated mitochondrial phenotypes in Drp1(-/17) expressing cells, in contrast to smaller, more circular mitochondria observed following expression of Drp1(16/17) (Fig. 3E,F; Appendix

Fig. S3). Notably, expression of Drp1(-/17) led to an overall higher number and density of cristae per mitochondria (Fig. 3E,F). These observations suggest that high expression of Drp1(-/17) leads to the dampening of mitochondrial fission and a more fused mitochondrial network. We speculate that this is due to decreased association of Drp1(-/17) with mitochondrial fission puncta and increased localization to microtubules. Moreover, after exposure to the pro-fission stimulus FCCP Drp1(-/17) did not associate with mitochondrial fission puncta, unlike Drp1(16/17). Instead, Drp1(-/17) continued to exhibit a high association with the microtubule network and mitochondria maintained their elongated mitochondrial morphology (Fig. EV3). This suggests that cells expressing Drp1(-/17) have reduced response to pro-fission stimuli. These findings are significant, as Drp1 splice variants display clear differences in localization and function as fission proteins, yet past studies have primarily investigated the function of Drp1 in cancer cells by overexpressing the Drp1(16/17) transcript that includes both exons 16 and 17. Thus, we next investigated the functional consequences of Drp1(-/17) expression in ovarian cancer cells.

## Expression of Drp1(-/17) increases oxygen consumption and alters tumor cell metabolism

Due to their continuous membranes and matrix lumens, fused mitochondrial networks can facilitate better diffusion of molecules, including ADP, and reducing equivalents NADH and FADH2, necessary for oxidative phosphorylation (Li et al, 2017; Skulachev, 2001). In addition, a more ordered cristae architecture improves electron transport chain super-complex assembly, and these structural features have been associated with optimal mitochondrial respiration (Cogliati et al, 2016; Cogliati et al, 2013; Varanita et al, 2015). Given that Drp1(-/17) splice variant expression leads to enhanced mitochondrial networks and cristae numbers, the effects on mitochondrial respiration and cellular metabolism were assessed. Using extracellular flux analysis, significant increases in basal oxygen consumption rates (OCR), ATP-dependent OCR and spare respiratory reserve were observed between Drp1(-/17) and Drp1(16/17) expressing OVCA433 cells (Fig. 4A,B). An increase in OCR was similarly observed in SKOV3 cells expressing Drp1(-/17) (Fig. EV4A). There were no differences in the expression of mitochondrial proteins COX-I (complex IV), and SDH-A (Complex II) following expression of either Drp1 variant suggesting that changes in mitochondrial activity are likely not attributable to changes in mitochondrial biogenesis (Fig. EV4B). In addition, no

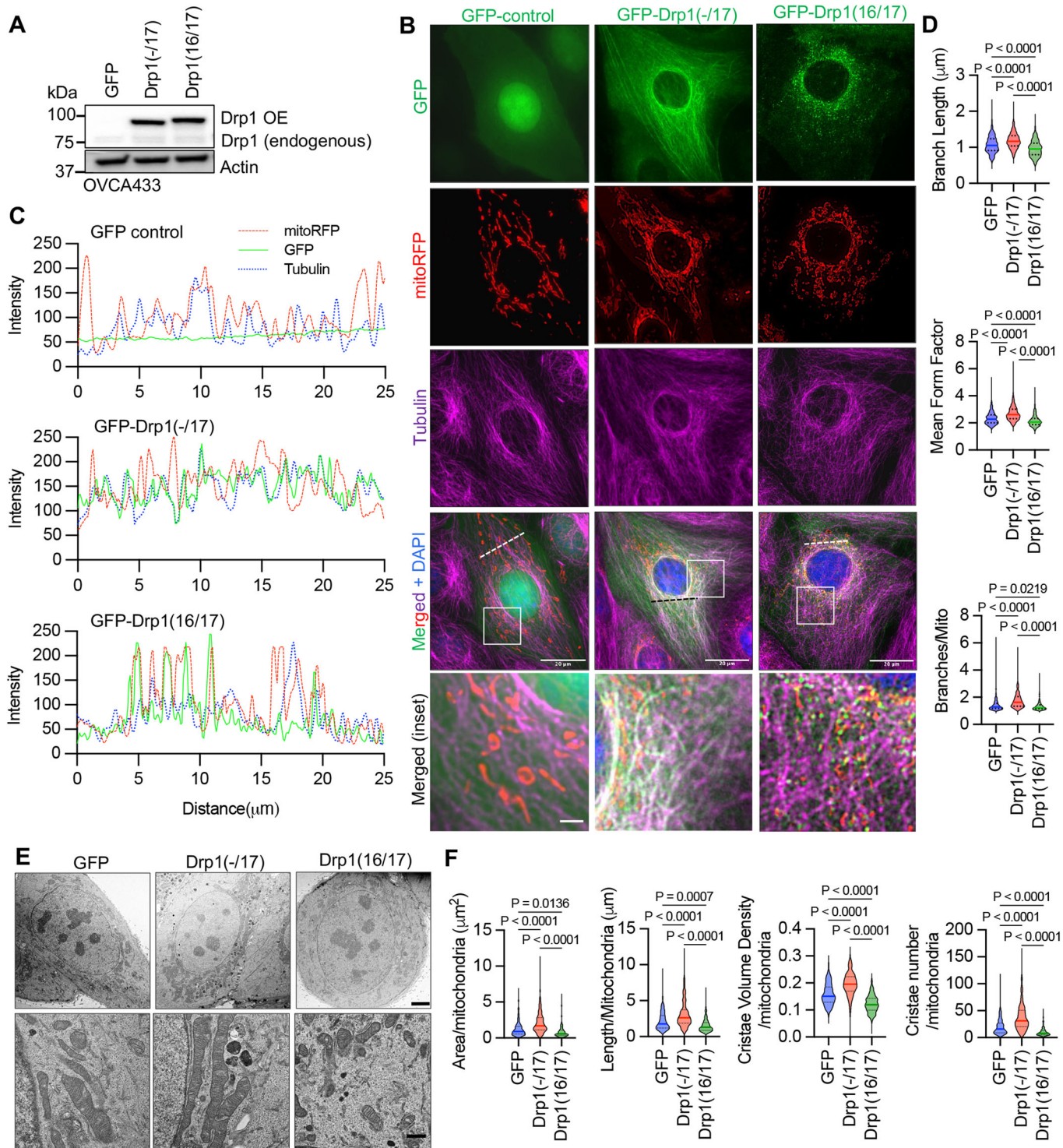

consistent changes in mitochondrial membrane potential (Fig. EV4C) or significant changes in MitoSOX fluorescence (Fig. EV4D), an indicator of mitochondrial oxidant production, were associated with the expression of either Drp1 variant in the cell lines tested.

Compared to Drp1(16/17) or GFP controls, untargeted metabolomics (Fig. 4C; Appendix Fig. S4) showed that cells expressing

Drp1(-/17) also display an increase in metabolites linked to glycolysis and the pentose phosphate pathway. Related to the observed increases in lactate levels, an increase in extracellular acidification rate was also observed, although this was not significant in OVCA433 cells (Fig. EV4E). Select metabolites necessary for de novo purine and pyrimidine synthesis, such ribulose-5-phosphate, glycine, and aspartate, and most TCA cycle metabolites were also elevated following

**Figure 3. Drp1(-/17) displays decreased association with mitochondria, and its expression increases the mitochondrial length and cristae density relative to Drp1(16/17).**

(A) Western blot analysis of Drp1 expression following overexpression (OE) of GFP vector control, GFP-Drp1(-/17) or GFP-Drp1(16/17) in OVCA433 cells. (B) Representative epifluorescence images of mitochondrial morphology and Drp1 distribution in OVCA433 cells. (Green: GFP or GFP-tagged Drp1, Red: mito-RFP to label mitochondria, Magenta: anti-Tubulin antibody, Blue: DAPI). Drp1(-/17) shows a distinct pattern of co-localization with Tubulin, while Drp1(16/17) displays localization to mitochondrial fission puncta. Scale bar: 20 μm, scale bar inset 2 μm. (C) Representative histograms of fluorescence intensity (from dotted line in panel B images) illustrate that Drp1(-/17) (green) is more closely aligned with Tubulin (blue) and less so with mitochondria (red). In contrast, GFP-Drp1(16/17) fluorescence peaks coincide with mitochondrial (red) peaks, reflective of association with mitochondrial fission puncta. (D) Drp1(-/17) expressing cells display elongated and branched mitochondrial networks compared to cells expressing Drp1(16/17). Quantification of mitochondrial morphology based on mito-RFP labeling by three independent descriptors (mitochondria analyzer, ImageJ; GFP control $n = 498$ cells, Drp1(-/17) $n = 568$ cells, Drp1(16/17) $n = 553$ cells; median + IQR, one-way ANOVA mean form factor $P < 0.0001$; one-way ANOVA branch length $P < 0.0001$; one-way ANOVA Branches/mito $P < 0.0001$. Tukey's post test $P$ values are shown). (E) Representative TEM images of OVCA433 cells demonstrate that Drp1(-/17) expressing cells have more fused mitochondria with greater cristae organization and volume compared to cells expressing Drp1(-1/7) or GFP control. Scale bar: 4 μm (upper panels) and 800 nm (lower panels). (F) Quantification of mitochondrial morphology and cristae from TEM images (GFP control $n = 156$ cells, Drp1(-/17) $n = 160$ cells, Drp1(16/17) $n = 157$ cells; median + IQR, one-way ANOVA area/mitochondria $P < 0.0001$; one-way ANOVA cristae volume density $P < 0.0001$; one-way ANOVA length/mitochondria $P < 0.0001$; one-way ANOVA cristae number/mitochondria $P < 0.0001$. Tukey's post test $P$ values are shown). Source data are available online for this figure.

overexpression of Drp1(-/17) relative to GFP control and Drp1(16/17) (Fig. 4C; Appendix Fig. S4). These changes were accompanied by elevated total NAD(H) levels and a decrease in the NAD + /NADH ratio (Figs. 4D and EV4F), which may be contributing to the reducing equivalents necessary for increased respiration seen in Drp1(-/17) cells. Collectively, these findings suggest that enhanced Drp1(-/17) expression leads to an energetic phenotype and alters the metabolism of ovarian cancer cells compared to cells expressing Drp1(16/17).

## Drp1 splice variant expression differentially affects tumorigenicity

*DNM1L* gene amplification has been reported to correlate with cell cycle gene expression and poor patient outcomes in chemoresistant and recurrent ovarian cancer cases (Tanwar et al, 2016). However, previous work did not account for the expression of specific Drp1 splice variants, and it is thus unknown which Drp1 transcript variant is specifically associated with this pro-tumorigenic phenotype. Considering that high Drp1(-/17) relative to low Drp1(16/17) expression is associated with lower overall patient survival (Fig. 2F), we further assessed the impact of Drp1 variable domain splice variant expression on tumor cell behavior. Drp1(-/17) expression increased the proliferation rate of both SKOV3 and OVCA433 cells, compared to GFP control and Drp1(16/17) (Fig. 5A). Conversely, Drp1(16/17) expression significantly decreased clonogenic survival (Fig. 5B). The most significant difference was observed in cellular migration, where cells expressing Drp1(-/17) exhibited increased migration compared to GFP and Drp1(16/17) expressing cells (Fig. 5C). An intraperitoneal tumor xenograft model demonstrated that expression of either Drp1(-/17) or Drp1(16/17) influences the location and degree of tumor burden in the peritoneal cavity (Fig. 5D–H). Cells expressing Drp1(16/17) displayed delayed onset of tumor growth and the lowest overall tumor burden within the peritoneal cavity (Fig. 5D–H), correlating with our observations in cell culture. Mice injected with GFP control expressing SKOV3 cells reached the endpoint first (Fig. 5E), with tumor burden largely localized to the peritoneal wall near the IP injection site (Fig. 5F,H). In contrast, mice injected with Drp1(-/17) cells developed tumors primarily in the omentum, a major site of ovarian cancer metastatic spread in the abdominal cavity (Fig. 5G,H). Omental tumors also differed morphologically, with cells expressing Drp1(-/17) forming a greater number of smaller and denser multi-lobular lesions

along the omentum (Fig. 5H iii,iv). Taken together, these data show for the first time that Drp1 splice variants differentially influence proliferation, migration, and in vivo growth of cancer cells, and that expression of Drp1(-/17) is advantageous to omental tumor progression.

## Drp1(-/17) protects cells against chemotherapy-induced apoptosis

Previous findings suggest the involvement of mitochondrial fission in the initiation of apoptosis (Arnoult et al, 2005; Frank et al, 2001; Karbowski et al, 2002), and thus decreased fission has been proposed as a mechanism of apoptosis resistance in cancer cells. Interestingly, Drp1(-/17) expression was previously shown to abrogate staurosporine-mediated cell death of astrocytes in cell culture (Strack et al, 2013). We thus sought to test if high expression of Drp1(-/17) and the concomitant increase in fused mitochondria may be beneficial to tumor cells when challenged with cisplatin or paclitaxel treatment (Fig. 6A). Expression of Drp1(-/17) led to a statistically significant increase in IC50 values for both compounds in OVCA433 and SKOV3 cells relative to GFP controls (Fig. 6A). Conversely, Drp1(16/17) expression enhanced sensitivity to both compounds, which was particularly evident with Cisplatin (Fig. 6A,B). Notably, cells with Drp1(16/17) expression exhibited the highest caspase 3/7 activity in response to both agents, while Drp1(-/17) expression significantly abrogated caspase activity relative to GFP and Drp1(16/17) expressing cells (Fig. 6B). These data suggest that Drp1(16/17) mediated mitochondrial fission enhances apoptosis of ovarian cancer cells, and that Drp1(-/17), which abrogates mitochondrial fission, protects cells from apoptosis. To determine if Drp1 splice variants differ in activation by post-translational modifications in response to chemotherapeutic agents, phosphorylation of Serine 616 was investigated. Although predominantly observed in OVCA433 cells, Drp1(16/17) was more susceptible to phosphorylation at S616 upon treatment with both cisplatin and paclitaxel, while no change in phosphorylation was observed in Drp1(-/17) in response to these agents (Appendix Fig. S5). Finally, the effects of Cisplatin on in vivo tumor growth were tested using a subcutaneous xenograft model (Fig. 6C). In the saline-treated groups Drp1(-/17) expression significantly increased final tumor volume compared to both GFP and Drp1(16/17) expressing SKOV3 cells when mice were euthanized at the same endpoint (Fig. 6D). Similar to observations in the omentum, subcutaneous

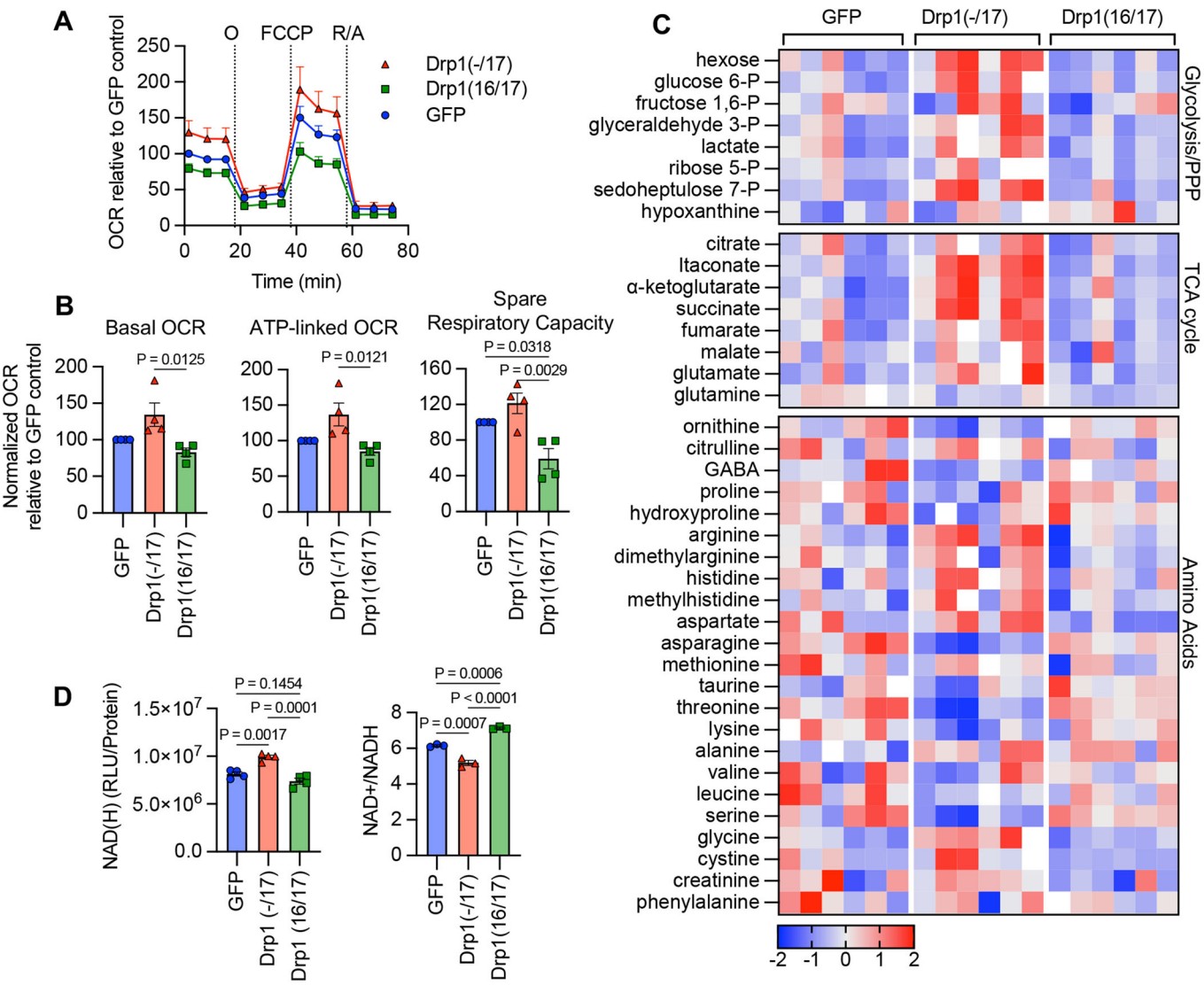

**Figure 4. Expression of Drp1(-/17) splice variant increases mitochondrial respiration and TCA cycle metabolites.**

(A) Expression of Drp1(-/17) increases oxygen consumption rates (OCR) in OVCA433 cells as assessed by Seahorse extracellular flux analysis and the mitochondrial stress test (O: oligomycin A, R/A: rotenone/antimycin A; OCR is normalized to cell viability and expressed relative to GFP control, mean ± SEM of four biological replicates each derived from the average of 2–4 technical repeats). (B) Basal OCR, ATP-linked OCR, and spare respiratory capacity are increased in OVCA433 cells expressing Drp1(-/17) compared to Drp1(16/17). Data were expressed relative to GFP control (mean ± SEM of four biological replicates, each derived from the average of 2–4 technical repeats, one-way ANOVA basal OCR $P = 0.0144$; ATP-linked OCR $P = 0.0131$; spare respiratory capacity $P = 0.0034$; Tukey's post test $P$ values shown). (C) Relative metabolite content of OVCA433 cells stably expressing GFP control, GFP-Drp1(-/17) or GFP-Drp1(16/17) as assessed by untargeted LC-HRMS ($n = 6$, heatmap reflects z-scores of analyte/ISTD peak area values). (D) Total NAD(H) levels are increased in response to Drp1(-/17) expression relative to OVCA433 cells expressing GFP control or Drp1(16/17), while the ratio of NAD + /NADH is significantly decreased (NAD(H): mean ± SEM of four biological replicates each derived from the average of three technical repeats, one-way ANOVA $P = 0.0002$; NAD + /NADH: mean ± SEM of four biological replicates each derived from the average of three technical repeats one-way ANOVA $P < 0.0001$; Tukey's post test $P$ values shown). Source data are available online for this figure.

tumors from Drp1(-/17) expressing cells were multi-lobular. In addition, mice injected with Drp1(-/17) expressing cells developed lymph node lesions (Fig. 6E), again suggesting that expression of Drp1(-/17) enhances the metastatic behavior of cells. In the cisplatin-treated groups, none of the GFP and Drp1(-/17) expressing tumors responded to cisplatin treatment and all progressed after cisplatin removal (day 40). On the contrary, 4/10 Drp1(16/17) expressing tumors either completely responded or did not reach a tumor volume threshold of 200mm³ at the study endpoint (Fig. 6C,F), validating cell

culture studies that Drp1(16/17) expression leads to higher chemo-sensitivity. Although final Drp1(-/17) tumor weights at the endpoint were lower than GFP controls in the cisplatin-treated groups, Drp1(-/17) tumors were multi-lobular and were the only group to develop lymph node lesions with cisplatin treatment (Appendix Fig. S6). Altogether, these data demonstrate that the loss of exon 16 through alternative splicing influences the function of Drp1 in tumor cells, leading to changes in metastatic and chemoresistance phenotypes.

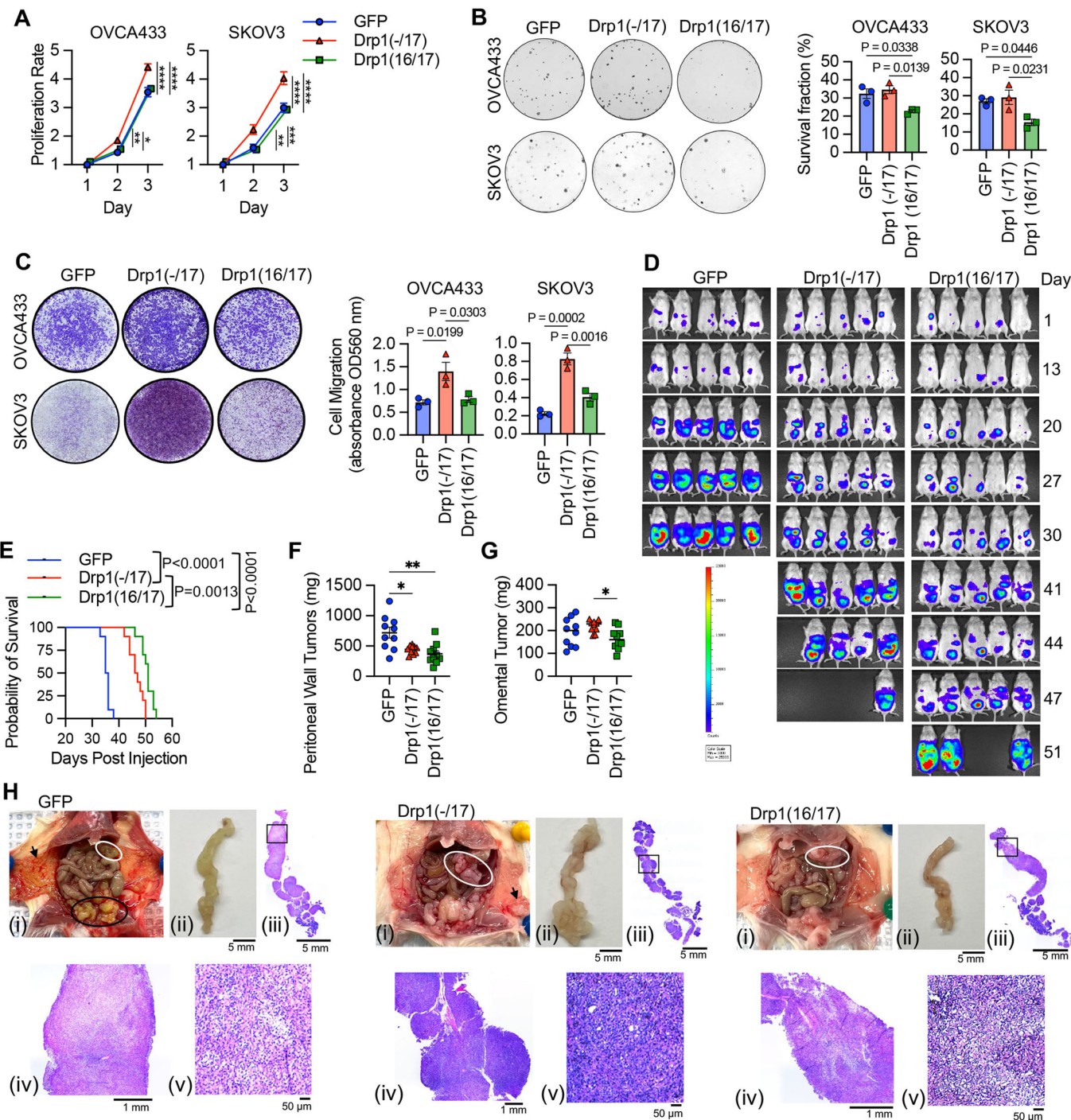

## Manipulation of endogenous Drp1 splice variant ratios can dictate the pro-tumorigenic function of Drp1(-/17)

The data above clearly show that expression of Drp1(-/17), the splice variant lacking exon 16, has significantly different effects on mitochondrial morphology and function, tumor cell behavior, and chemosensitivity compared to expression of the full-length Drp1(16/17). To rule out that these observations are due to overexpression of the recombinant protein, we sought to assess

whether altering the endogenous ratios of Drp1 splice variants could elicit similar effects. To achieve this, splice variant-specific siRNAs targeting each of the four endogenous variable domain Drp1/*DNM1L* splice variants were designed (Figs. 7A and EV5A). This validated that the major Drp1 protein variants identified by western blotting in OVCA433 cells (Fig. 1B) are Drp1(-/17) and Drp1 (16/17) (Appendix Fig. S7). SKOV3 cells were used as a model for subsequent experiments, as this cell line displays relatively equal expression of the four variable domain Drp1

Figure 5.   Compared to Drp1(16/17), expression of Drp1(-/17) promotes proliferation and migration, maintains clonogenic survival and drives omental tumor burden of ovarian cancer cells.

(A) Drp1(-/17) expression increases proliferation of OVCA433 and SKOV3 cells. Cell proliferation was assessed by FluoReporter dsDNA quantification, and proliferation rate expressed as increase in the cell density relative to day 1 (mean ± SEM of three biological replicates each derived from the average of four technical repeats, OVCA433: two-way ANOVA group factor variance $P < 0.0001$, Tukey's post test *$P = 0.0034$, **$P = 0.0044$, and ****$P < 0.0001$; SKOV3: two-way ANOVA group factor variance $p < 0.0001$, Tukey's post test **$P = 0.0049$, ***$P = 0.0007$, and ****$P < 0.0001$). (B) Drp1(16/17) expression lowers single-cell clonogenic survival in both OVCA433 and SKOV3 cells. Cells (100/well) were seeded onto six-well plates and stained with crystal violet after 7–10 days in culture. Colonies were quantified using ImageJ. Representative image of one technical replicate shown (mean ± SEM of three biological replicates each derived from the average of six technical repeats, one-way ANOVA OVCA433 $P = 0.0128$; SKOV3 $P = 0.0201$. Tukey's post test $P$ values shown). (C) Drp1(-/17) expressing cells are more migratory than Drp1(16/17) or GFP control cells. Cell migration was assessed using the Boyden chamber transwell assay. Images are representative of three independent assays (mean ± SEM of three biological replicates each derived from the average of two technical repeats, one-way ANOVA OVCA433 $P = 0.0155$; SKOV3 $P = 0.0003$. Tukey's post test $P$ values shown). (D) Peritoneal tumor burden was monitored using bioluminescence imaging at indicated days after NSG mice were injected with $1 \times 10^6$ SKOV3 cells expressing GFP control, Drp1(-/17) or Drp1(16/17) (5 representative mice/group shown). (E) Survival of mice injected IP with SKOV3 cells expressing GFP control, Drp1(-/17) or Drp1(16/17) ($n = 10$ mice/group, Log-rank Mantel-Cox test $P$ values shown; median survival GFP: 33.5 days, Drp1(-/17): 46.5 days, Drp1(16/17): 51 days). (F) Peritoneal tumor weight was measured after removal at necropsy when mice met endpoint ($n = 10$, median, Kruskal–Wallis $P = 0.0046$, uncorrected Dunn's test *$P = 0.0448$, **$P = 0.0011$). (G) Omental tumor weight was measured after removal at necropsy when mice met endpoint ($n = 10$, median, Kruskal–Wallis $P = 0.0576$, uncorrected Dunn's test *$P = 0.017$). (H) Representative image of mice with tumors from SKOV3 cells expressing GFP control, Drp1(-/17) or Drp1(16/17). (i) GFP-expressing cells primarily formed peritoneal tumors near the injection site (Black outline) and peritoneal wall (arrow), while Drp1(-/17) expressing cells primarily formed tumors in the omentum (white outline). Gross omental tumors (ii) and sectioned omenta followed by H& E staining (iii–v) demonstrate that Drp1(-/17) expressing cells form multiple nodular tumors along the omentum. Source data are available online for this figure.

transcript variants (Fig. 7A), thus allowing us to test the effects of tuning the ratios of their endogenous expression using siRNAs. Specific knock-down of individual splice variants transcripts was validated by RT-PCR flanking the variable domain (Fig. 7A) and at the protein level (Fig. 7B). While it was more difficult to resolve Drp1(16/17) and Drp1(16/-) on SDS-PAGE due to their close molecular weight individual knockdowns of Drp1 protein variants could be distinctly visualized (Fig. 7B). A positive control siRNA designed to target Exon 15 led to decreased protein expression of all variants (Fig. 7A,B, lane 2). The knock-down of all variants (siDrp1-total) resulted in fused mitochondrial morphology as expected, and as demonstrated by previous studies targeting Drp1 by RNAi (Fonseca et al, 2019; Mopert et al, 2009; Otera et al, 2010) (Fig. 7C,D). When targeting specific Drp1 splice variants with siRNA, we observed varying degrees of impact on mitochondrial networks (Figs. 7C,D and EV5B,C). Drp1(16/17) knockdown most closely replicated the highly elongated mitochondrial morphology and increased branching seen with total Drp1 knockdown (siDrp1-total, Fig. 7C,D). In contrast, with knockdown of Drp1(-/17) these effects were muted (Fig. 7C,D), again highlighting that Drp1(-/17) contributes less to mitochondrial fission than Drp1(16/17) (Fig. 7E). The combination knockdown of Drp1(-/16) and Drp1(−/−), which essentially enriches for higher but equal expression of Drp1(16/17) and Drp1(-/17), led to an intermediary increase in mitochondrial length and branching (Fig. 7C,D). This suggests that the endogenous ratio of expression between the Drp1(16/17) and Drp1(-/17) splice variants is critical in determining the overall fission activity of Drp1. Collectively, these results confirm our overexpression data (Fig. 3), suggesting that the Drp1(16/17) variant plays a more active role in mitochondrial fission, while Drp1(-/17) contributes less to mitochondrial fission (Fig. 7E).

Importantly, the functional consequences of altered Drp1 splice variants expression on mitochondrial function could similarly be recapitulated using siRNA to target endogenous Drp1 transcripts (Fig. 8). By recombinant expression, we showed that Drp1(16/17) was detrimental to mitochondrial respiratory function while Drp1(-/17) expression improved OCR (Figs. 4 and EV4). In agreement with this, knock-down of Drp1(16/17) improved

respiration (Fig. 8B) while siRNA targeting of Drp1(-/17) decreased both basal and ATP-linked oxygen consumption rate (OCR) (Fig. 8C). No significant changes in mitochondrial function were observed upon combined knockdown of Drp1(16/-) and Drp1(−/−) which enriched for equal expression of Drp1(16/17) and Drp1(-/17) (Fig. 8D). These data again suggest that an imbalance of Drp1(16/17) and Drp1(-/17) expression is necessary to drive changes in mitochondrial function.

Subsequently, we investigated the changes in cellular proliferation and migration upon perturbation of endogenous Drp1 variants. Total knockdown of all Drp1 transcripts (siDrp1-total) drastically reduced both cellular proliferation and migration (Fig. 8E,F). Increased cellular proliferation was only seen when Drp1(16/17) was knocked down (Fig. 8E), or when specific enrichment of Drp1(-/17) was achieved by a combined knockdown of the other three variable domain variants (siDrp1(16/17),(-/16),(−/−); Fig. 8Aii,F). These conditions recapitulate the effects of Drp1(-/17) overexpression (Fig. 5A,B) and mimic the low Drp1(16/17):high Drp1(-/17) expression observed in the TCGA patient cohort that is marked by poor overall patient survival (Fig. 2F). Similarly, cell migration was significantly enhanced following Drp1(16/17) knockdown and decreased after Drp1(-/17) knockdown (Fig. 8G). Finally, in contrast to a loss of Drp1(16/17), the knockdown of Drp1(-/17) significantly increased apoptosis in response to cisplatin and paclitaxel treatment (Fig. 8H), which confirms that Drp1(-/17) protects cells from apoptosis, while expression of Drp1(16/17) enhances chemosensitivity. Thus, shifting endogenous expression to a high Drp1(-/17):low Drp1(16/17) ratio using siRNA validates the pro-proliferative, migratory, and chemoresistance phenotypes observed by Drp1(-/17) variant overexpression (Figs. 5, 6). The above findings highlight the importance of the expression ratio between the Drp1(16/17) and Drp1(-/17) variants. This is demonstrated by the observation that a combined knockdown of Drp1(-/16) and Drp1(−/−), reflecting an equal enrichment of expression of both Drp1(16/17) and Drp1(-/17), led to relatively minor changes in mitochondrial morphology, with no significant impact on mitochondrial function, cellular proliferation, migration or

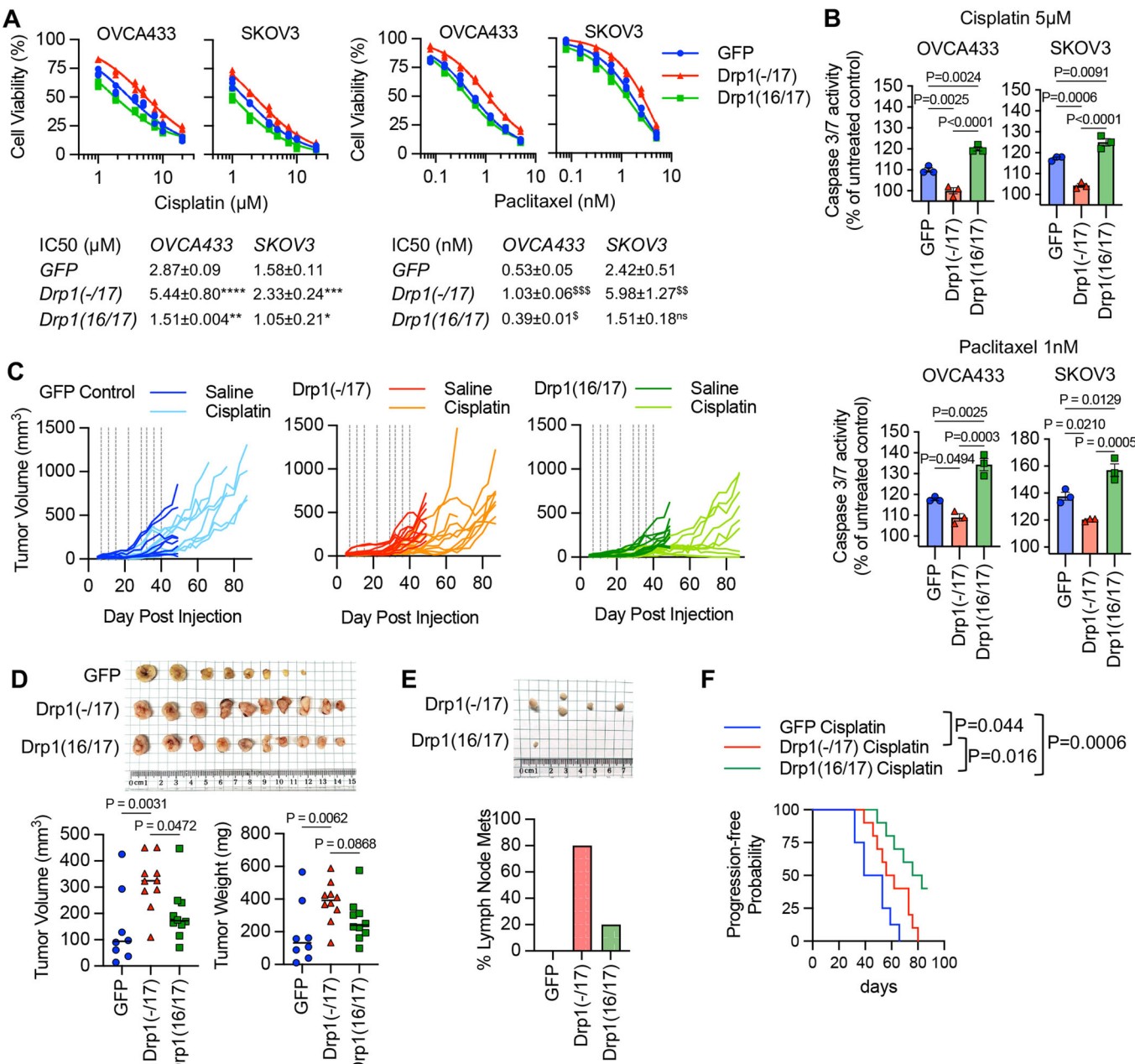

chemosensitivity. Taken together, these data emphasize that changes in the ratio of Drp1 splice variant expression can have profound effects on mitochondrial morphology and metabolism that have consequences on tumor cell function.

## Discussion

To our knowledge, this is the first detailed description of Drp1/ *DNM1L* transcript variants and their functional significance in a pathophysiological setting. While Drp1 has been extensively studied in various diseases, including multiple cancers (Banerjee et al, 2022; Kashatus et al, 2015; Lima et al, 2018; Tanwar et al, 2016; Xie et al, 2015), the investigation of different Drp1 isoforms

arising from alternative splicing has remained limited, with few studies conducting direct comparisons (Chen et al, 2000; Ciesla et al, 2021; Itoh et al, 2018; Strack et al, 2013; Uo et al, 2009). Moreover, prior research has often neglected to specify which Drp1 isoform was the subject of study. It is assumed that most prior works utilized plasmid that contain all exons of the variable domain (i.e., Drp1(16/17)) to overexpress recombinant Drp1, or that knock-down strategies targeted the expression of all Drp1 variants. Our findings demonstrate that alternative splicing of exon 16 is an important feature of ovarian cancer, that this is associated with poor patient outcomes, and that alternative splicing of exon 16 results in the expression of *DNM1L*/Drp1 proteins with distinct roles related to the regulation of mitochondrial form and function, which impacts pro-tumorigenic behavior in culture and in vivo.

**Figure 6. Drp1 splice variant expression affects chemosensitivity of ovarian cancer cells.**

(A) Expression of Drp1(-/17) decreases sensitivity to cisplatin and paclitaxel. Dose-response curves were derived from cell viability assays (FluoReporter dsDNA quantification) of OVCA433 and SKOV3 cells expressing GFP vector control, Drp1(-/17) and Drp1(16/17) in response to cisplatin and paclitaxel treatment (72 h; IC50s calculated from three independent experiments, mean ± SD, one-way ANOVA, Dunnet's post test comparison to GFP control, P values for Cisplatin: *P = 0.0306, **P = 0.0202, ***P = 0.0061, ****P = 0.001; P values for Paclitaxel: $P = 0.0179, $$P = 0.0028, $$$P < 0.0001). (B) Cells expressing Drp1(-/17) display abrogated apoptosis in response to cisplatin (5 μM) or paclitaxel (1 nM) treatment after 24 h, as assessed using Caspase-Glo 3/7 assay (mean ± SEM of three biological replicates each derived from the average of 3–4 technical repeats, one-way ANOVA cisplatin OVCA433 P = < 0.0001, SKOV3 P = < 0.0001; paclitaxel OVCA433 P = 0.0003, SKOV3 P = 0.0006. Tukey's post test P values shown). (C) Drp1(16/17) expression leads to Cisplatin sensitivity of SKOV3 cells in vivo. Subcutaneous tumor growth of SKOV3 cells expressing GFP control, Drp1(-/17) or Drp1(16/17) was monitored following injected into female CrTac:NCr-Foxn1$^{nu}$ mice (tumor volume of individual tumors shown, 2 tumors per mouse, n = 4–5 mice). Saline or cisplatin (5 mg/kg) was administered IP at indicated days (vertical lines on graphs). 4/10 tumors in the Dpr1(16/17) group responded to cisplatin treatment, while all GFP and Drp1(-/17) expressing tumors progressed with treatment. (D) Final tumor volume and weight of SKOV3 subcutaneous tumors from saline-treated groups. All mice were euthanized at the same endpoint (day 49; n = 8, GFP; n = 10, Drp1(-/17); n = 10, Drp1(16/17); median shown, tumor volume Kruskal–Wallis P = 0.0211, uncorrected Dunn's test P values shown; tumor weight Kruskal–Wallis P = 0.0102, uncorrected Dunn's test P values shown). (E) Drp1(-/17) expressing SKOV3 cells develop lymph node metastases in subcutaneous tumor model. Lymph node metastases were resected in the saline-treated groups at day 49 (cisplatin-treated group see Appendix Fig. S6B). Graph shows the percentage of mice with lymph node metastases. (F) Progression-free probability of cisplatin-treated mice demonstrates that Drp1(16/17) promotes cisplatin sensitivity of SKOV3 cells. Tumor progression was determined as tumor burden reached volume >200 mm³ (n = 8 tumors, GFP; n = 10, Drp1(-/17); n = 10 Drp1(16/17); Log-rank Mantel-Cox test P values shown, median probability GFP: 46 days, Drp1(-/17): 59 days, Drp1(16/17): 79.5 days; final tumor weight at endpoint see Appendix Fig. S6A). Source data are available online for this figure.

Thus, our work strongly supports that future studies focused on Drp1 should take the expression and function of *DNM1L*/Drp1 splice variants into consideration.

According to TCGA data the *DNM1L* gene is amplified in >50% serous ovarian cancer cases (5% high level amplification, 46% low level gain), while only 7% of cases show heterozygous loss (Appendix Fig. S8A). Published work has correlated *DNM1L* amplification with enhanced cell cycle gene expression and poor survival in chemoresistant and recurrent cancer samples (Tanwar et al, 2016; Tsuyoshi et al, 2020). Notably, previous analyses of mRNA expression did not discern levels of specific Drp1 splice variants, leaving it uncertain which transcripts are associated with cell cycle gene expression, prognosis, and chemoresistance. We found that ovarian cancer cells derived from patient ascites, as well as TCGA ovarian cancer specimens express high levels of the transcript lacking exon 16 (Drp1(-/17); Figs.1, 2). Strikingly, high Drp1(-/17) expression relative to Drp1(16/17) correlates with worse overall patient survival (Fig. 2). The identification that a specific Drp1 splice variant is linked to unfavorable patient outcome underscores the clinical impact of Drp1 splice variant expression in cancer for the first time.

We established that the two major ovarian cancer Drp1 splice variants, Drp1(-/17) and Drp1(16/17) have distinct localization and effects on mitochondrial morphology and function (Figs. 3, 4). Drp1(-/17) was previously reported to localize to microtubules (Strack et al, 2013). Similarly, we observed that Drp1(-/17) was associated more frequently with microtubules rather than mitochondria compared to Drp1(16/17). This likely explains why Drp1(-/17) expression shifts mitochondrial morphology towards a fused state. Importantly, through splice-site specific siRNA-mediated knockdown of individual Drp1 variants, we provide the first validation of the fusion phenotype associated with Drp1(-/17) at endogenous expression levels. The variant-specific knockdowns also emphasize the significance of stoichiometric expression of the Drp1 splice variants in fine-tuning regulation of mitochondrial morphology, possibly representing a novel mechanism exploited by cancer cells to manipulate their mitochondrial dynamics and subsequently mitochondrial function (Figs. 7, 8).

Decreased fission, and a consequentially more fused mitochondrial network has been shown to enhance electron transport chain super-complex assembly and to improve mitochondrial respiratory function (Cogliati et al, 2016; Gao et al, 2020). While prior work in Kras mutant pancreas cancers showed that complete knockdown of Drp1-mediated fission leads to abrogated hexokinase activity, decreased glycolysis and proliferation, and loss of mitochondrial function (Nagdas et al, 2019), we find that a more nuanced decrease in fission associated with Drp1(-/17) expression enhances mitochondrial respiratory function. Drp1(-/17) and is associated with a more compact cristae arrangement (Figs. 3, 4). This potentially enables cancer cells to thrive under stress when heightened mitochondrial function is necessary, and may explain the advantage Drp1(-/17) expression confers for in vivo tumor growth, compared to Drp1(16/17) (Figs. 5, 6). Stress-induced mitochondrial fusion has been implicated with stress resistance and to promote cell survival (Li et al, 2017; Tondera et al, 2009; van der Bliek, 2009). Similarly, the fused networks resulting from elevated Drp1(-/17) expression likely allow ovarian cancer cells to bolster mitochondria robustness and maintain metabolic efficiency. Cells expressing Drp1(-/17) display a more active metabolic phenotype, characterized by elevated respiration, glycolysis, and TCA cycle metabolites (Fig. 4). Numerous studies have emphasized the significance of dynamically adapting mitochondrial morphology to drive metabolic flexibility and regulate tumor cell survival (Anderson et al, 2014; Grieco et al, 2020; Li et al, 2017; Ngo et al, 2023; Sessions et al, 2022; Tondera et al, 2009; van der Bliek, 2009). The preference for metabolic fuels is also closely associated with mitochondrial architecture (Alan and Scorrano, 2022; Liesa and Shirihai, 2013; Ngo et al, 2023). The reliance of tumor cells on manipulating their mitochondrial respiration and function is highlighted by numerous recent studies (Bellance et al, 2009; Dar et al, 2017). Ovarian cancer stem cells and chemoresistant cells exhibit a remarkably adaptable metabolic phenotype, capable of switching between glycolysis and oxidative phosphorylation depending on which pathway confers a selective growth advantage and chemoresistance (Anderson et al, 2014; Dar et al, 2017; Dier et al, 2014; Ghoneum et al, 2020; Li et al, 2017; Pasto et al, 2014; Yang et al, 2014). Moreover, metabolic alterations are essential to sustain unbridled growth in cancer cells, with increased ATP synthesis and a shift towards de novo macromolecule biosynthesis. While the increased mitochondrial network and improved cristae architecture observed in Drp1 (-/17)

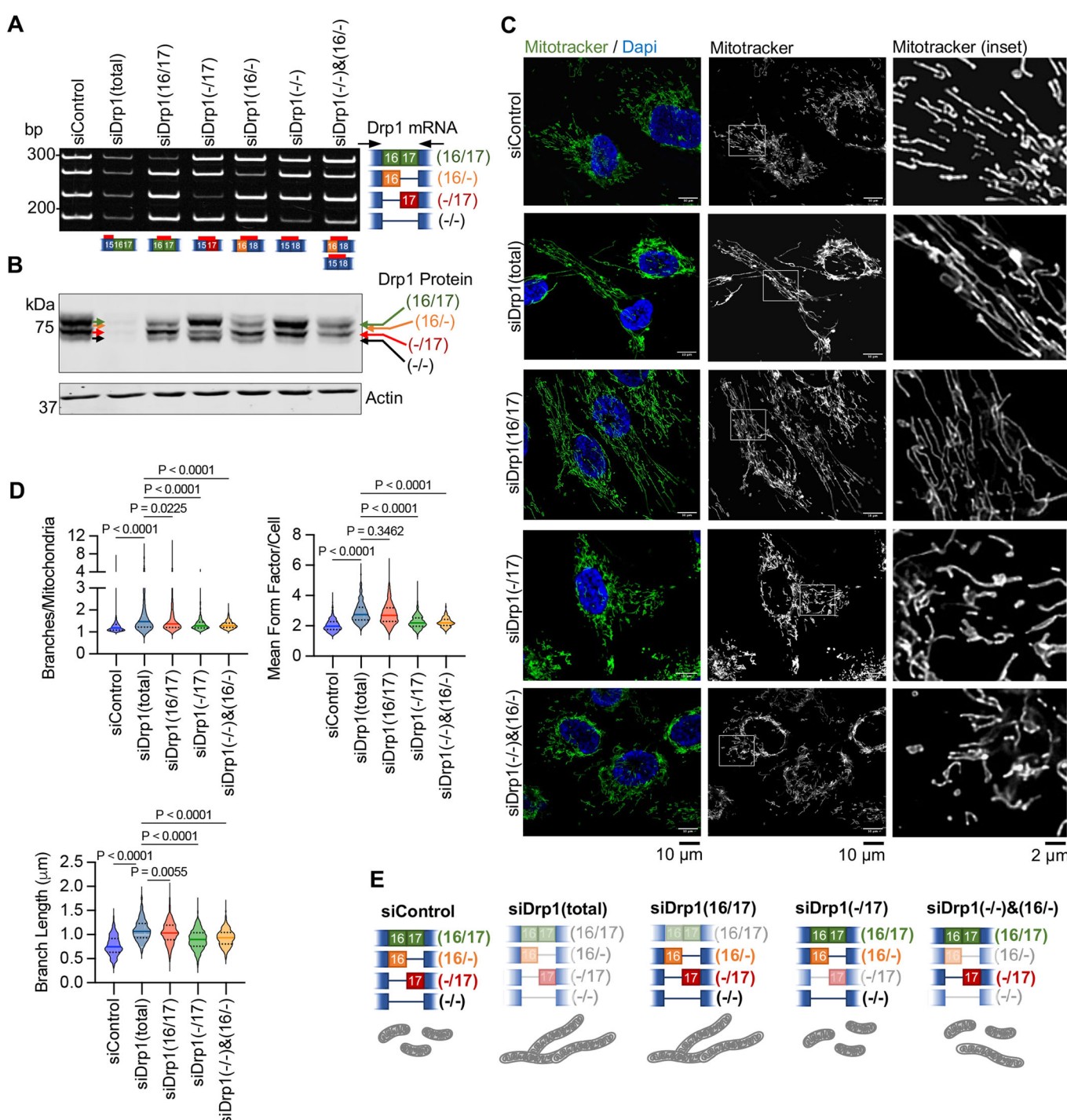

potentially contributes to ETC complex assembly and efficiency in oxidative phosphorylation, the increase in TCA cycle metabolites is also suggestive of an enhanced TCA cycle flux, which could be driven by the need for NADH reducing equivalents, as demonstrated by increased levels of NADH in Drp1(-/17) cells. It remains to be determined if mitochondrial architecture also contributes to the activity or efficiency of TCA cycle enzymes, although it is known that substrate availability is enhanced by a more fused mitochondrial network (Cogliati et al, 2016; Skulachev, 2001).

Prior metabolic profiling of ovarian cancer cells and tumors derived from patients has revealed discernible differences in purine and pyrimidine metabolism, glycerolipid metabolism, and energy metabolism (Denkert et al, 2006; Fong et al, 2011; Yang et al, 2014). Interestingly, akin to these observations, Drp1(-/17) expression correlated with heightened levels of amino acids like glutamine, glycine, and aspartate, all crucial components for purine and pyrimidine metabolism (Fig. 4). Furthermore, the decreased serine levels and increased glycine levels associated with Drp1(-/17)

Figure 7. Specific knock-down of endogenous Drp1 splice variants using siRNA and effects on mitochondrial morphology.

(A) RT-PCR demonstrating variant-specific knock-down of endogenous Drp1 using splice variant-specific siRNA in SKOV3 cells. One representative gel from three independent replicates shown. (B) Changes in Drp1 protein levels following splice variant-specific siRNA-mediated knock down of Drp1 protein in SKOV3 cells were visualized after resolving proteins on 7.5% SDS-PAGE followed by western blotting (antibody: ab184247). One representative blot from 3 independent replicates is shown. (C) Representative epifluorescence images of mitochondrial morphology upon splice variant-specific siRNA Drp1 knockdown in SKOV3 cells. (Green: mitotracker green, Blue: DAPI). The disruption of endogenous Drp1 splice variant expression differentially modifies mitochondrial dynamics. siDrp1(16/17) most closely replicates the elongated mitochondrial morphology observed following knock-down of all Drp1 variants (siDrp1-total). Scale bar: 10 μm. (D) Quantification of mitochondrial morphology represented by three independent descriptors using ImageJ mitochondria analyzer (siControl $n = 560$ cells, siDrp1(total) $n = 334$, siDrp1(16/17) $n = 630$, siDrp1(-/17) $n = 655$, siDrp1($-/-$)&(16/-) $n = 555$; median + IQR, one-way ANOVA mean form factor $P < 0.0001$; one-way ANOVA branch length $P < 0.0001$; one-way ANOVA branches/mito $P < 0.0001$. Tukey's post test was performed to assess differences between groups and $P$ values for comparisons to siDrp1(total) are shown). (E) Schematic representation of siRNA-mediated knock-down of Drp1 splice variants and effects on endogenous splice variant expression and mitochondrial morphology. Source data are available online for this figure.

expression suggest enhanced catabolism of the nonessential amino acid serine through one-carbon metabolism, essential for purine nucleotide synthesis. It remains to be determined if the expression of Drp1(-/17) and the concomitant improved mitochondrial function thus grants ovarian cancer cells greater metabolic flexibility and confers survival advantages in the face of shifting nutrient availability, or if an expression of specific Drp1 splice variants can lead to metabolic vulnerabilities that could be harnessed for therapeutic intervention.

We provide the first demonstration that splice variants of Drp1 have distinct impacts on cell proliferation, migration, sensitivity to chemotherapeutics, and in vivo tumor growth (Figs. 5, 6). Without considering Drp1 splice variants, Drp1 has been extensively studied as a regulator of cell death and proliferation. Drp1-mediated fission has been shown to be crucial for mitosis in cancer cells (Kashatus et al, 2015; Taguchi et al, 2007). Yet it is known that during the G1-S transition, mitochondrial networks fuse, which aids in cyclin E-mediated G1-S transition, and inhibition of Drp1-mediated fission can induce the proliferation of quiescent cells (Mitra et al, 2009). Similarly, elevated mitochondrial fusion has been identified as a feature of ovarian cancer neoplastic stem cells, priming them for self-renewal and proliferation (Spurlock et al, 2021). Interestingly, the microtubule localization of Drp1(-/17) has been associated with phosphorylation by cyclin-dependent kinases, where Cdk1 leads to Drp1(-/17) microtubule dissociation, while CDK5 mediates microtubule association of Drp1(-/17) during interphase, suggesting regulation of the subcellular distribution of different splice variants during the cell cycle (Strack et al, 2013), which requires further investigation in the context of cancer and differential Drp1 variant expression.

A significant finding from this work is that Drp1 splice variants alter the chemosensitivity of ovarian cancer cells (Figs. 6, 8H). Improved mitochondrial function and increased oxidative phosphorylation, as seen in ovarian cancer cells with Drp1(-/17) splice variant expression, are established phenotypes of chemoresistance development in tumor cells, including ovarian cancer (Farge et al, 2017; Zampieri et al, 2020). Moreover, since mitochondrial fission is an essential process during apoptosis (Arnoult et al, 2005; Caino et al, 2015; Farrand et al, 2013; Frank et al, 2001; Kong et al, 2014b; Thomas and Jacobson, 2012), the increased fused mitochondrial networks in Drp1(-/17) expressing cells may further account for the observed resistance to apoptosis upon exposure to agents like cisplatin and paclitaxel. As such, previous studies have reported higher abundance of tubular and elongated mitochondria in

chemoresistant ovarian cancer cells (Kong et al, 2014a; Zou et al, 2021). We propose that heightened resistance to chemotherapy, coupled with the survival advantages conferred by improved mitochondrial function, contribute to the poor outcome observed in patients with high Drp1(-/17) tumor expression (Fig. 2).

Given the unique association of Drp1(-/17) with microtubules, the target site of taxanes, further investigations are warranted to ascertain whether the heightened chemoresistance conferred by Drp1(-/17) is attributed to reduced mitochondrial fission or its interaction with microtubules. The interaction of Drp1 with microtubules was previously shown to be driven by direct, electrostatic interactions between the conserved basic residues in Drp1 exon 17 (Arg566/567) and the acidic N-termini of α/β-tubulin. The existence of exon 16 in the Drp1(16/17) sequence is hypothesized to impede this interaction, possibly by physically obscuring or neutralizing the positive charge of the neighboring microtubule-binding domain (Strack et al, 2013). Although further evidence is needed, such as an assessment of heterodimer formation of Drp1 splice variants, it is possible that high expression of Drp1(-/17) could act as a dominant negative isoform by sequesters other splice variants away from mitochondria to microtubules. Future work is needed to elucidate potential extra-mitochondrial functions of Drp1(-/17) and how these could be additionally contributing not only to abrogated fission activity, but other cellular functions that play a role in driving the enhanced tumorigenic features of cells predominantly expressing this variant.

Shortcomings of this work that require further investigation include our lack of understanding as to why and how exon 16 is specifically spliced out in ovarian cancer cells. Cancer cells, in general, are known for their alterations in RNA splicing and processing (Ciesla et al, 2021; Song et al, 2019; Tien et al, 2017), and it remains to be determined if specific alterations in the RNA splicing machinery give rise to higher expression of Drp1(-/17). As such, aberrant expression of RNA splice factors such as SRSF3 is associated with ovarian cancer (He et al, 2011; Zhu et al, 2018), and we find that Drp1(-/17) is more strongly expressed in tumors with documented *CDK12* mutations (Appendix Fig. S8B), a cyclin-dependent kinase that has been associated with regulation of the spliceosome (Tien et al, 2017).

In summary, our study sheds light on the pathophysiological importance of Drp1 variant expression and their ability to modify mitochondrial fission and fusion dynamics as a novel mechanism underlying ovarian cancer cell plasticity. This study also emphasizes the necessity of expanding our comprehension of these Drp1 splice variants beyond the scope of cancer, and for their

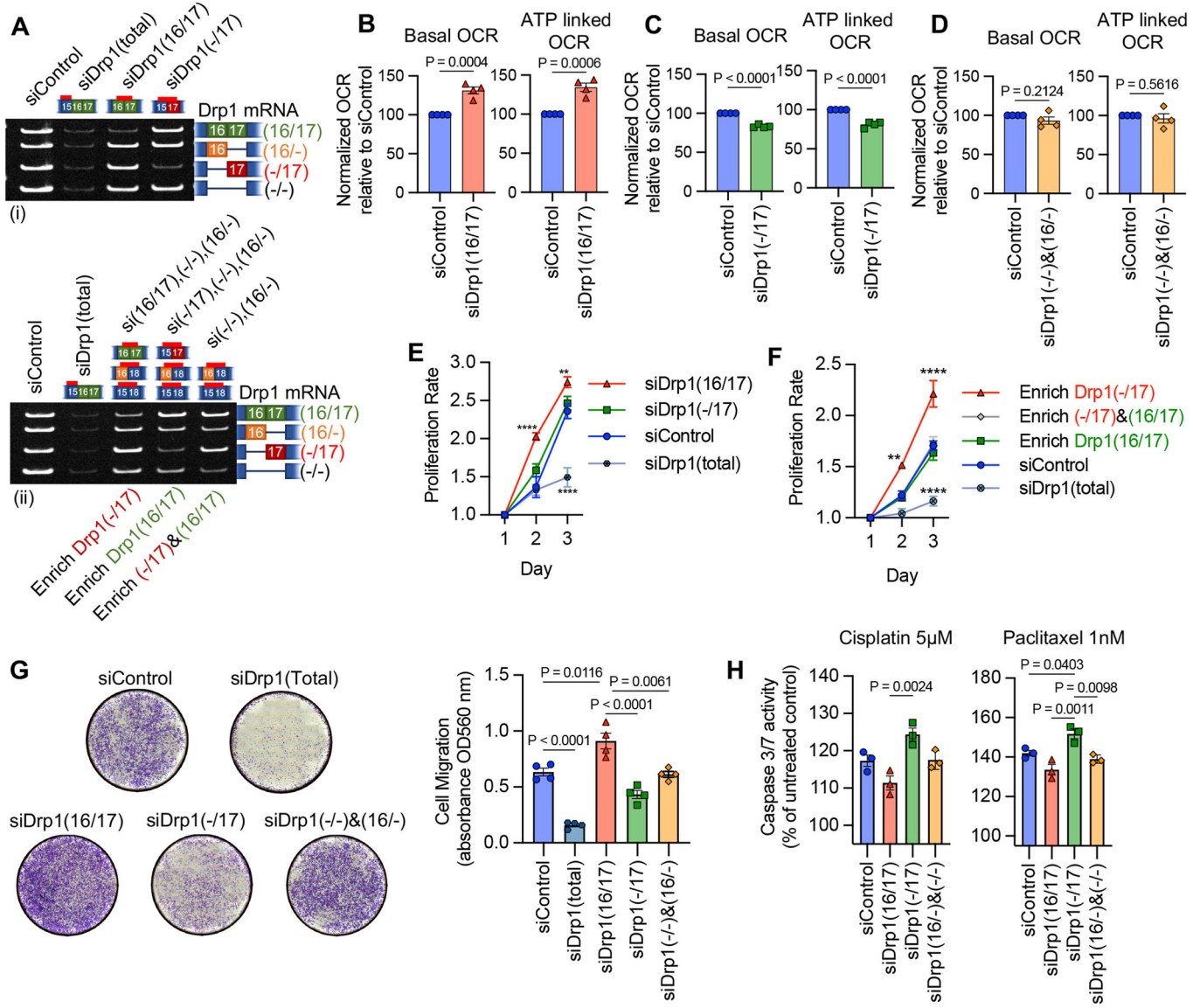

**Figure 8. Targeted knock-down of endogenous Drp1 splice variants in SKOV3 cells differentially affect mitochondrial respiration, proliferation, and migration.**

(A) RT-PCR demonstrating knock-down of Drp1 using single splice variant-specific siRNA or combination of siRNAs to enrich for specific Drp1 splice variant expression. One representative gel from independent replicates shown. (B) Basal and ATP-dependent oxygen consumption rates (OCR) improved upon siRNA-mediated knockdown of Drp1(16/17) splice variant, which increases the Drp1(-/17):Drp1(16/17) ratio. Mitochondrial respiration was assessed using Seahorse extracellular flux analysis and the mitochondrial stress test. Data expressed relative to siControl, mean ± SEM of four biological replicates each derived from the average of three technical repeats, unpaired *t*-test *P* values shown. (C) Conversely, specific knockdown of Drp1(-/17) decreased basal and ATP OCR (mean ± SEM of four biological replicates each derived from the average of three technical repeats, unpaired *t*-test *P* values shown). (D) Mitochondrial respiration remains unchanged when enriching for equal levels of Drp1(-/17) and Drp1(16/17) expression by a combination of Drp1(16/-) and (Drp1(−/−) knock-down. (mean ± SEM of four biological replicates each derived from the average of three technical repeats, unpaired *t*-test *P* values shown). (E) Single variant knock-down of Drp1(16/17) increases proliferation rate of SKOV3 cells relative to siControl (*n* = 3, mean ± SEM, two-way ANOVA group factor variance *P* < 0.0001, Tukey's post test *P* values for comparisons to siControl shown, **P = 0.0086, ****P < 0.0001). (F) Drp1(-/17) variant enrichment with combination knock-down increases the proliferation rate of SKOV3 cells relative to siControl. Cell proliferation was assessed by FluoReporter dsDNA quantification and proliferation rate expressed as an increase in the cell density relative to day 1 (*n* = 3, mean ± SEM, two-way ANOVA group factor variance *P* < 0.0001, Tukey's post test *P* values for comparisons to siControl shown, **P = 0.0012, ****P < 0.0001). (G) Endogenous Drp1 splice variants differentially affect cell migration in SKOV3 cells. Knock-down of Drp1(16/17) increases migration, while Drp1(-/17) knock-down reduces cell migration relative to siControl cells. After siRNA-mediated knock-down, cell migration was assessed using a Boyden chamber transwell assay. Images are representative of four independent assays (*n* = 4, mean ± SEM, one-way ANOVA ****P < 0.0001. Tukey's multiple comparison post test *P* values shown). (H) Cells with single variant knock-down of Drp1(16/17) have reduced apoptotic response to cisplatin (5 nM) or paclitaxel (1 nM) treatment compared to knock-down of Drp1(-/17). Apoptosis was assessed using Caspase-Glo 3/7 assay after 24 h of treatment (mean ± SEM of three biological replicates each derived from the average of three technical repeats, one-way ANOVA cisplatin *P* = 0.0040; paclitaxel *P* = 0.0016, Tukey's multiple comparison post test *P* values shown). Source data are available online for this figure.

consideration when investigating the function of Drp1 in different (patho)physiological settings.

# Methods

## Cell lines and cell culture

Cell lines were generously provided by the following investigators: OVCA433, OVCA420 Dr. Susan K. Murphy; HeyA8, Dr. Katherine Aird; FT282, Dr. Ronny Drapkin. ES-2, TOV-21-G, and OVCAR3 cells were purchased from the American Type Culture Collection (ATCC, CRL-1978, HTB-161). OVCA433, OVCA420, SKOV3, and HeyA8 cells were cultured in RPMI 1640 medium (Corning,10-040-CV) supplemented with 10% fetal bovine serum, FBS (Avantor® Seradigm, 1500-500). OVCA433 and SKOV3 cells under selection were maintained in fully supplemented growth media with 5 ug/ml Puromycin (Gibco™, A1113803). OVCAR3 cells were cultured in RPMI 1640 medium supplemented with 0.01 mg/ml bovine insulin (Fisher Scientific,50-608-896), and 10% FBS ES-2 cells were cultured in Modified McCoy's 5a Medium (Corning,10-050-CV) with 10% FBS. TOV-21-G cells were cultured in a 1:1 mix of MCDB 105 (Sigma-Aldrich,117-500) containing 1.5 g/L sodium bicarbonate (Gibco™, 25080094) and Medium 199 (Corning, 10-060-CV) containing 2.2 g/L sodium bicarbonate, with 15% FBS. FT282 cells were cultured in 50% DMEM and 50% Ham's F-12 medium (Corning,10-090-CV) supplemented with 2% FBS. All cells were maintained at 37 °C with 5% $CO_2$. Cell lines were routinely tested for Mycoplasma contamination using the EZ-PCR™ Mycoplasma Detection Kit (Captivate Bio, 20-700-20). Cell line authentication was carried out by STR genotyping (Labcorp).

## Patient ascites-derived EOC cells

Epithelial ovarian cancer (EOC) cells were isolated from malignant ascites of ovarian cancer patients treated at the Women's Cancer Care clinic (Albany, NY) and the Penn State Hershey College of Medicine Division of Gynecologic Oncology, with approval granted from the State University of New York at Albany IRB (Protocol # 14-E-058-01) and the Penn State College of Medicine IRB (Protocol # STUDY00004648), respectively. Informed consent was sought from all study participants from the patient's physician prior to de-identified specimens being sent to the laboratory for analysis. Histological subtype and staging of EOC samples are provided in Fig. 2C. Following procurement of ascites EOCs were immediately isolated and cultured as previously described (Shepherd et al, 2006), and maintained in culture at 37 °C, 5% $CO_2$ in MCDB/M199 medium supplemented with 10% FBS and penicillin/streptomycin.

## Matched tumor specimens

Archival matched specimens of normal fallopian tube or ovary, ovarian tumor, and omental tumor from high-grade serous ovarian cancer patients were obtained through an honest broker from the ProMark biospecimens bank at the University of Pittsburgh Magee Womens Research Institute, with approval granted from the University of Pittsburgh IRB (Protocol # STUDY21050194). Histological subtype and staging of tumor specimens are provided in shown in Appendix Fig. S2. Acquisition and use of patient specimens conformed to the principles set out in the WMA declaration of Helsinki and the Department of Health and Human Services Belmont Report.

## 5′/3′ Rapid amplification of cDNA ends (RACE)

3′ and 5′RACE reactions were carried out using the SMARTer 3′5′ RACE kit (Takara), essentially as recommended by the manufacturer. 5′RACE was carried out with the Universal Primer A Mix (UPM) and the *DNM1L* specific antisense primer, positioned in Exon 12: 5′ GTTCCACACAGCGGAGGCTGGGC 3′. 3′RACE was carried out using *DNM1L* primer spanning the exon junction of exon 6/7: 5′ GATTACGCCAAGCTTTGCCAGGAATGACCA AGGTGCCTGT-3′, followed by nested PCR on the PCR product using *DNM1L* specific primers in exon 10/11: 5′ GATTACGCC AAGCTTACTTCGGAGCTATGCGGTGGTGCT-3′. RACE PCR products were resolved on a 10% agarose gel, bands gel purified and cloned into the pRACE vector for subsequent sequencing of inserts. PolyA sites were annotated to the *DNM1L* gene using the NCBI genome browser tracks for PolyA sites and clusters, Polyadenylation sits from PolyA_DB (v.3.2), and the polyadenylations sites from the PolyASites at the University of Basel.

## Drp1/DNM1L variant RT-PCR

Total RNA from cells and tissue was isolated using the Direct-zolTM RNA Miniprep kit (Zymo Research, R2052). Prior to RNA purification, tissue was crushed under liquid Nitrogen, and 25–50 mg of tissue-powder lysed in 800 ml Ambion TRIzol Reagent (Invitrogen™,15596018) overnight at 4 °C. First-strand cDNA synthesis was performed using qScript cDNA Synthesis Kit (Quantabio, 95047) according to the manufacturer's instruction. RT-PCR was performed using diluted cDNA (1:5 in water) and PrimeStar DNA Polymerase (Takara, R010A) with primers and PCR conditions listed in Table 1. The amplified products, mixed with Gel Loading Buffer II (Thermo Scientific, AM8547), were separated on a PA-TBE gel and stained with GelStarTM Nucleic Acid Gel Stain (Lonza, 50535).

## siRNA-mediated knock-down

Cells were transfected with scramble non-targeting SMARTpool control (Dharmacon#D-001810-10-05) or single or combination of Drp1 variant-specific siRNA oligonucleotides against target sequence listed in Table 2 using Lipofectamine RNAiMAX (Invitrogen, 13778150). Forty-eight hours post-transfection cells were seeded for experiments. For each experiment, knock-down was confirmed by RT-PCR using Drp1 variable domain variant RT-PCR primers, as above.

## Immunoblotting

Cells were cultured to sub-confluency and lyzed in RIPA buffer (Thermo Scientific™, 89901) containing protease and phosphatase inhibitors (Thermo Scientific™, 78443). The protein supernatant was collected following 30 min rotation at 4 °C, followed by maximum speed (21,000 rcf) centrifugation for 30 min in a 1.5 ml tabletop centrifuge at 4 °C. Protein concentrations were measured using the Pierce BCA protein assay kit (Thermo Scientific™, 23225).

**Table 1. Primers and PCR cycle conditions for Drp1/DNM1L variant RT-PCR.**

| Primer | Sequence (S: sense; AS: antisense) | PCR cycle |
|---|---|---|
| Drp1 variable domain variants (Exons 16/17) | S: 5′-GGCAATTGAACTGGCTTATATCAACAC-3′<br>AS: 5′-TGGTTGGTTCTTGAACACCATCTCCAA-3′ | 98C-15s, (98C-10s, 70C-15s, 72C-20s) X32 |
| Drp1 ΔC truncated variants | S: 5′-GGCAATTGAACTGGCTTATATCAACAC-3′<br>AS: 5′-TAGATA CCACTACACAAACAGGTTCTT-3′ | 98C-15s, (98C-10s, 70C-15s, 72C-20s) X32-40 |
| Total Drp1 (Exons 1–2) | S: 5′-GTGGGCCCCGGCCCCATTCAT-3′<br>AS: 5′- CAGTACCTCTGGGAAGCAGGTCCCTCC-3′ | 95C-3m, (95C-10s, 68C-15s, 72C-20s) X32 |
| Actin | S: 5′-AACTGGGACGACATGGAG-3′<br>AS: 5′- TAGCACAGCCTGGATAGCAACGTA-3′ | 98C-15s, (98C-10s, 70C-15s, 72C-20s) X24 |

Following SDS-PAGE, proteins were transferred to PVDF membranes. Membranes were blocked for an hour in 5% non-fat milk (Bio-Rad,1706404)/TBS, 0.1% Tween20 (MilliporeSigma, 900-64-5) and were probed overnight at 4 °C in primary antibodies (Table 3). The next day blots were incubated for 1 h at room temperature with horseradish peroxidase (HRP)-conjugated secondary antibodies and were developed using SuperSignal™ West Femto Maximum Sensitivity Substrate Femto (Thermo Scientific™, 34096).

## TCGA data analysis

RNAseq bam files for the TCGA serous ovarian carcinoma cohort ($n = 379$, all cases regardless of grade and stage) were downloaded from dbGaP and stored on a secure server according to dbGaP protocols. The samtools 1.9.0 software (Li et al, 2009) was used to convert each bam file to two fastq files corresponding to the paired-end reads after randomizing the order of the reads with "samtools collate." The salmon 1.3.0 software (Patro et al, 2017) was then applied to quantify genome-wide transcript abundances based on an index created from a custom reference fasta file, as described below, that incorporates sequences from splice variants and alternate polyadenylation forms of DNML1. The --seqBias, --gcBias, and --validateMapping options for the salmon quant command were utilized. The salmon index was created by following the steps outlined at https://combine-lab.github.io/alevin-tutorial/2019/selective-alignment/. First, a decoys.txt file was created using the GRCh38 primary assembly fasta file. Next, a custom version of the gencode.v22.pc_transcripts.fa file was made by removing all transcripts corresponding to the DNML1 gene and replacing them with sequences corresponding to the alternative splice and alternatively polyadenylated transcripts of *DNML1*. Concatenating this custom transcriptome fasta file with the GRCh38 primary assembly fasta file yielded the gentrome fasta file that, along with the decoys.txt file, was used to produce the salmon index. Overall survival data were obtained using cBioportal.

## Drp1 plasmids and subcloning

Rat Drp1(-/17) and Drp1(16/17) cloned in pEGFP-C1 plasmids were kindly provided by Dr. Stefan Strack, University of Iowa (Strack et al, 2013). Drp1 coding sequence were subcloned into pLenti-CMV-MCS-GFP-SV-puro (Addgene, 73582) with a N-terminal GFP tag and sequenced to confirm successful cloning of Drp1(-/17) and Drp1(16/17) plasmids. The plasmids were transfected in 293-FT cells for expression and lentiviral particle production. OVCA433 and SKOV3 cells were infected and transduced with the GFP vector control, GFP-Drp1(-/17) and

GFP-Drp1(16/17) virus, selected for expression using 5 ug/ml Puromycin for 1–2 weeks and sorted for GFP expression by flow cytometry to generate stable Drp1 overexpressing cells.

## Immunofluorescence and analysis of mitochondrial morphology

Prior to imaging, mitochondria were labeled in cells by transduction with a pLV-mitoDsRed virus (Addgene, Plasmid 44386) harvested post expression in 293-FT cells and lentiviral particle production. For imaging, cells were seeded at 60–70% confluency in Chambered Cell Culture slides (Falcon, 08-774-25). The next day, cells were washed once with 1X PBS (Corning, 21-040-CV) and fixed in 4% formaldehyde, made fresh by diluting 16% PFA solution (BTC BeanTown Chemical,30525-89-4) in 1X PBS for 10 min at room temperature. For imaging FCCP treatment, cells were incubated with 1 µM FCCP(Sigma) in media for 30 min prior to fixation. Post fixation, cells were rinsed twice with 1X PBS and permeabilized for 10–15 min using 0.2% Triton™ X-100 (Fisher Scientific, BP151500) in 1X PBS with gentle rocking. Followed by two more PBS rinses prior to 1 h incubation with SuperBlock™ Blocking Buffer (Thermo Scientific™,37515) for blocking non-specific antibody binding. Cells were stained with primary antibody against Tubulin (Abcam, ab6160) at 1:1000 dilution in SuperBlock™ blocking buffer for either 1 h 30 min at room temperature or overnight at 4 °C. After incubation, cells were washed for three subsequent 10 min washes with 1X PBS with gently rocking. Cells were incubated with 1:1000 dilution of secondary rat antibody conjugated with Alexa Fluor® 647 (abcam, 150167) in SuperBlock™ Blocking Buffer at room temperature for 30 min. After, washed with 1X PBS three times,10 min each to remove any residual antibody. Slides were mounted in ProLong™ Gold Antifade Mountant with DNA Stain DAPI (Invitrogen, P36935) and dried overnight in the dark at room temperature. For mitochondrial imaging of SKOV3 post siRNA transfection, cells were seeded on 35 mm glass bottom dishes (MatTek, P35GC-1.5-14-C). The next day cells were stained using MitoTracker™ Green FM (Invitrogen™, M7514) following the manufacturer's recommendations. Briefly, cells were incubated for 12–15 min in a pre-warmed culture medium containing 250 nM MitoTracker™ Green FM and Hoechst 33342 (Thermo Scientific™,62249). After, cells were washed thrice with 1X PBS and imaged in pre-warmed 1X HBSS (Corning, 21-023-CV). Z-stacks were taken with Leica Thunder Imager using 63X oil immersion objective and subjected to inbuilt thunder de-convolution. The mitochondrial network morphology to look at the elongation and fragmentation was performed on 2D Z-stack projections in image J using the mitochondria-analyzer plugin (https://github.com/AhsenChaudhry/Mitochondria-Analyzer). Briefly, the images were converted into

**Table 2. DNM1L/Drp1 splice variant-specific siRNA target sequences.**

| Drp1 siRNA | Target Sequence - 5′-3′ | Target location |
|---|---|---|
| siDrp1(total) | GAGAAACAGGCTAGCCAGAGAATTACCTTCA | Exon 15 |
| siDrp1(16/17) | GAGGCTGATGGCAAGTTAATTCAGGACAGCA | Exon 16/17 junction |
| siDrp1(-/17) | CTGTATCACGAGACAAGTTAATTCAGGACAG | Exon 15/17 junction |
| siDrp1(16/-) | GAGGCTGATGGCAAGGTTGCATCTGGAGGTG | Exon 16/18 junction |
| siDrp1(−/−) | CTGTATCACGAGACAAGGTTGCATCTGGAGG | Exon 15/18 junction |

**Table 3. Antibodies used.**

| Antibody | Manufacturer | Cat # | Dilution |
|---|---|---|---|
| Drp1 | Abcam | 184247 | 1:1000 |
| Drp1 | EMD Millipore | ABT155 | 1:1000 |
| Phospho-DRP1 (Ser616) | Cell Signaling | 3455 | 1:1000 |
| MitoBiogenesis™ Western Blot Cocktail | Abcam | 123545 | 1:250 |
| β-actin (9f3) | Thermo Scientific | AM4302 | 1:1000 |
| β-tubulin (AC-15) | Cell Signaling | 2128 | 1:1000 |
| GAPDH (0411) | Santa Cruz | sc-47724 | 1:1000 |
| Vinculin | Sigma-Aldrich | Aldrich V9131 | 1:1000 |
| Amersham ECL HRP conjugated rabbit IgG | Cytiva | NA934 | 1:10,000 |
| Amersham ECL HRP conjugated mouse IgG | Cytiva | NA931 | 1:10,000 |

binary, and the threshold was adjusted to detect the mitochondrial network before performing the 2D per-cell mitochondrial network analysis. A total of at least 150 or more cells were analyzed to get the morphological network parameters with 30 cells or more. analyzed for each biological replicate over at least three or more biological replicates. Representative images shown were adjusted in brightness and contrast for better visualization.

## TEM

Cells were fixed in cold 2.5% glutaraldehyde (25% glutaraldehyde stock EM grade, Polysciences, 111-30-8) in 0.01 M PBS (Fisher), pH 7.3. Samples were rinsed in PBS, post-fixed in 1% osmium tetroxide (Electron Microscopy Sciences) with 1% potassium ferricyanide, (Fisher), rinsed in PBS, dehydrated through a graded series of ethanol, and embedded in Poly/Bed® 812 (Luft formulations). Semi-thin (300 nm) sections were cut on a Reichart Ultracut (Leica Microsystems), stained with 0.5% Toluidine Blue O (Fisher) in 1% sodium borate (Fisher), and examined under the light microscope. Ultrathin sections (65 nm) were stained with 2% uranyl acetate (Electron Microscopy Science) and Reynold's lead citrate (lead nitrate, sodium citrate, and sodium hydroxide, Fisher) and examined on JEOL 1400 Plus transmission electron microscope with a side mount AMT 2k digital camera (Advanced Microscopy Techniques). For morphometric analysis of mitochondria by transmission electron microscopy, mitochondrial area, mitochondrial length (major axis), cristae number, and cristae volume density per mitochondria was quantified and analyzed as described

(Lam et al, 2021) using ImageJ. At least 40–50 mitochondria per biological replicate over three experimental replicates were analyzed.

## TMRE protocol

OVCA433 and SKOV3 were seeded into 96-well plates at a density of 1000 cells per well and incubated overnight in a cell culture incubator at 37 °C and 5% $CO_2$. The following day, half the sample wells were treated with 10 µM FCCP and incubated for 30 min in a cell culture incubator. All the sample wells were then treated with 100 nM TMRE (Thermo Fisher Scientific, T669), and incubated for 30 min at 37 °C in a cell culture incubator. All samples were subsequently washed twice with 1X PBS, and fluorescence measurements were taken with a Synergy HT microplate reader (BioTek) with an excitation wavelength of 530 nm and emission wavelength of 590 nm.

## MitoSOX mitochondrial superoxide indicator

About 500,000 cells were stained with 5 µM MitoSOX Red mitochondrial superoxide indicator (Invitrogen, M36008) in 1 ml HBSS (Corning, 21-023-CV) for 30 min at 37 °C. Following incubation, cells were washed with HBSS, and fluorescence was measured using flow cytometry as per the manufacturer's protocol. Unstained cells were used as a negative control, while cells treated with 50 µM Antimycin A were used as a positive control for flow cytometry.

## Bioenergetic analysis of oxygen consumption rate (OCR) and extracellular acidification rate (ECAR)

The Agilent Seahorse XFp Metabolic Analyzer (Agilent, model S7802A) was used to assess mitochondrial respiration of OVCA433 and SKOV3 as described previously for attached cells (Javed et al, 2022). Briefly, prior to the start of the experiment, cells were evenly seeded and cultured overnight in a Seahorse XFp cell culture plate (Agilent, 103022-100) at a density of 10,000 and 8000 cells/well for OVCA433 and SKOV3, respectively. The XFp sensor cartridge was hydrated in Agilent Seahorse XF Calibrant (Agilent,103022-100) at 37 °C in a humidified incubator (non-CO₂) incubator overnight. On the day of the experiment, cell culture media was replaced with pre-warmed seahorse XF base RPMI media, pH 7.4 (Agilent,103576-100) supplemented with 1 mM sodium pyruvate (Agilent 103578-100), 2 mM glutamine (Agilent 103579-100), and 10 mM glucose (Agilent, 103577-100). Cells were then placed into a non-$CO_2$ humidified incubator at 37 °C for 60 min. Mitochondrial stress test reagents (pharmacological manipulators of mitochondrial respiratory chain proteins) were diluted in pre-warmed XF assay media to achieve the following final concentrations in the cell culture well: 1.5 µM Oligomycin

A (Sigma, 75351); 1 or 0.5 µM FCCP (Sigma, C2920) for OVCA433 and SKOV3, respectively; and 0.5 µM Antimycin A/Rotenone (Sigma, A8674,45656). Three basal rate measurements (3 min measurement time each) were taken prior to the injection of mitochondrial stress test reagents and three measurements of OCR/ECAR were obtained respectively following injection of compounds. Post-run, the cells were stained with crystal violet dye (0.05%) (Sigma-Aldrich, 229288) for seeding normalization. The dye was released from cells using 30% acetic acid and absorbance was measured at 590 nm, using GloMax Explorer (Promega) microplate reader.

## Metabolomics

About 1–2 million cells were seeded in 60 mm dishes per replicate ($n = 6$ for each condition) in normal cell growth media. The next day, cells were washed once with 1X PBS, and fresh media was added. After 24 h, metabolic quenching and polar metabolite pool extraction was performed by adding ice-cold 80% methanol (aqueous with 0.1% formic acid) at a ratio of 500 µL per 1e6 cells. An internal standard mix (10 µM final concentration) consisting of creatinine-d3, alanine-d3, taurine-d4, and lactate-d3 (Sigma-Aldrich) was added to the sample lysates. Samples are scraped into Eppendorf tubes on ice, homogenized using a 25 °C water bath sonicator, and the supernatant was then cleared of protein by centrifugation at $16,000 \times g$. Cleared supernatant (2 µL) was subjected to untargeted liquid chromatography high-resolution mass spectrometry (LC-HRMS) analysis. Briefly, samples were injected via a Thermo Vanquish UHPLC and separated over a reversed-phase Thermo HyperCarb porous graphite column ($2.1 \times 100$ mm, 3 µm particle size) maintained at 55 °C. For the 20-min LC gradient, the mobile phase consisted of the following: solvent A ($H_2O$ + 0.1% formic acid) and solvent B (ACN + 0.1% formic acid). The gradient was the following: 0–1 min 1% B, with an increase to 15% B over 5 min, following a second increase to 98% B over 5 min. The gradient was held at 98% B for five minutes before column equilibration at 1% B for 5 min. The high-resolution Thermo Fisher ID-X tribrid mass spectrometer was operated in both positive and negative ion mode, scanning in ddMS$^2$ mode (2 µscans) from 70 to 800 $m/z$ at 120,000 resolution with an AGC target of 2e5 for full scan, and 2e4 for MS$^2$ scans using HCD fragmentation at stepped collision energies of 15, 35, and 50. Source ionization settings included spray voltages of 3.0 and 2.4 kV for positive and negative mode, respectively. Source gas parameters were 35 sheath gas, 12 auxiliary gas at 320 °C, and eight sweep gas. Calibration was performed prior to analysis using the Pierce™ FlexMix Ion Calibration Solutions (Thermo Fisher Scientific). Integrated peak areas were then extracted manually using Quan Browser (Thermo Fisher Xcalibur ver. 2.7). Relative amounts are reported as a ratio of the analyte peak area/internal standard peak area and converted to z-scores for heatmap data presentation.

## NAD⁺ and NADH level determination

OVCA433 and SKOV3 cells were seeded into flat-bottom 96-well plates at a density of 10,000 and 8000, respectively, in a normal growth medium. After overnight incubation, total NAD+ and NADH levels and their ratios were determined using an NAD+/NADH-glo assay kit (Promega, G9071) as per the manufacturer's protocol. Following the

addition of NAD/NADH-GloTM Detection Reagent, samples were incubated for 1 h at room temperature and transferred to a white flat-bottom 96-well plate (BrandTech®, BRA-781605). Bioluminescence was measured with a GloMax Explorer (Promega) plate reader. Protein Concertation was determined by BCA and used for data normalization.

## Cell proliferation and viability

Equal number of cells were seeded into 96-well plates at density of 1000 and 500 cells/well for OVCA433 and SKOV3 respectively, in normal culture medium. For cell proliferation rate, cell numbers were analyzed for 3 days successively using FluoReporter™ Blue Fluorometric dsDNA Quantitation Kit (Invitrogen™, F2962) as per manufacturer's protocol, and fluorescence measurements were taken with a Victor X (PerkinElmer) microplate reader with an excitation wavelength of 360 nm and emission wavelength of 460 nm respectively. The proliferation rate was measured as the increase in the cell density relative to day 1. For cell viability in response to chemotherapeutic agents, cells were treated with indicated doses of Cisplatin (cis-diamineplatinum(II) dichloride) (Sigma) or Paclitaxel(Sigma). Following 72 h of drug treatment, cell viability was measured using FluoReporter™ Blue Fluorometric dsDNA Quantitation Kit and was expressed as percentage survival relative to non-treated cells.

## Caspase 3/7 activity assay

OVCA433 and SKOV3 cells were seeded into flat-bottom 96-well plates at a density of 7000 and 5000 cells per well, respectively. After overnight incubation, cells were treated with 5 µM Cisplatin (cis-diamineplatinum(II) dichloride) (Sigma) or 1 nM Paclitaxel (Sigma). After 24 h drug treatment, caspase 3/7 activity was measured using the Caspase-Glo® 3/7 Assay System- (Promega, G8091) per the manufacturer's protocol. Briefly, an equal volume of reagent was added to the samples and only media control. Following 1 h incubation in the dark at room temperature, samples were transferred to a white flat-bottom 96-well plate (BrandTech®, BRA-781605), and bioluminescence was measured with GloMax Explorer (Promega) plate reader.

## Clonogenicity assay

About 100 cells/well were seeded in a six-well plate for single-cell survival clonogenicity assay and cultured for 7–10 days under normal culture conditions. Clonogenicity was assessed by staining colonies with crystal violet (0.05%). Colonies were counted using ImageJ and data expressed as cellular survival fraction.

## Migration assay

For assessment of Transwell migration, cell culture inserts (Corning, 353097) with 8.0 µm Transparent PET Membrane, were each placed into a 24-well plate with 800 µL complete growth medium (with serum) added at the bottom of each well. 50,000 and 30,000 cells of OVCA433 and SKOV3, respectively, were seeded onto the top of the Transwell membrane of the insert in 150 µL serum free media. After 24 h, the Transwell inserts were removed and washed twice with 1X PBS. The Transwell membrane was fixed

and stained with Crystal violet (0.05%) for an hour. The inserts were washed three times with 1X PBS and the non-migrated cells were removed from the top of the membrane using dry cotton swabs. The inserts were dried overnight, and images were taken of the migrated cells at the bottom of the membrane using a Leica Thunder Imager with a colored K3C camera. The dye was released from cell inserts using 30% acetic acid and absorbance was measured using GloMax Explorer (Promega) microplate reader at 590 nm as a readout for cell migration.

## Tumor xenografts

Approval for animal studies was sought from the University of Pittsburgh IACUC prior to study commencement (approved protocol IS00020674). All animals were housed in barrier facilities for immunodeficient mice in the University of Pittsburgh Animal Faculty, and mouse husbandry and experiments were performed in accordance with the guidelines of the Laboratory Animal Ethics Committee of Pittsburgh University. Mice were randomized prior to tumor cell injection and drug treatment. For Intraperitoneal xenografts $2 \times 10^6$ SKOV3 cells expressing either GFP control, GFP-Drp1(-/17) and GFP-Drp1(16/17) were suspended in 0.2 ml PBS and injected IP into female Nod *scid* gamma mice (NSG, Jackson laboratory; $n = 10$, sample size based on pilot study). Luminescence imaging was carried out on days indicated using an IVIS luminescence imaging system 10 min after mice were injected with 10 µl/g of body weight 15 mg/mL in vivo grade D-Luciferin (PerkinElmer, 122799). Mice were sacrificed when reaching AAALAC-defined endpoints by $CO_2$ asphyxiation followed by cervical dislocation. At necropsy, organs and tumor mass were weighed, and tissues were fixed in 10% buffered formalin (Fisher Scientific, 5735), followed by paraffin embedding, sectioning, and H&E staining. For Subcutaneous (SC) tumor xenografts SKOV3 cells expressing GFP, Drp1(-/17) and Drp1(16/17) were counted ($5 \times 10^6$) and prepared as suspensions in 0.2 ml PBS for both right and left subcutaneous (flank) injections into 6–8 week old CrTac:NCr-*Foxn1^{nu}* female mice (Taconic) in groups of 4–5 mice with total 8–10 tumor site injections per group. Two mice were removed from the study prior to the endpoint due to illness unrelated to tumor burden. Mice were injected IP with Cisplatin (5 mg/kg body weight) or vehicle control (PBS) on days 7, 11, 15, 22, 29, 32, 36, and 40 post tumor cell injection. Subcutaneous tumor growth was monitored using caliper measurements twice weekly, and tumor volumes were calculated according to the formula V = ½ (length (longer diameter) × width (shorter diameter)$^2$). All vehicle-treated groups were euthanized on the same day (Day 49) to compare basal tumor growth rates. Cisplatin-treated mice continued to be monitored for chemotherapy response and progression-free survival determined by tumor sizes of less than 200 m$^3$. Mice were euthanized by $CO_2$ asphyxiation followed by cervical dislocation when tumors reached >1500 mm$^3$ or other AAALAC endpoints were met. At endpoint, the xenografted tumors were resected and weighed.

## Data and statistical analysis

All data presented were from at least three biological replicates and represented as mean ± standard error of the mean. Analyses of images was blinded and randomized when possible. Unless otherwise indicated, statistical data analysis was carried out using GraphPad Prism Software (10.0.2), with appropriate analyses chosen based on experimental design, as stated in figure legends.

## Data availability

This study includes no data deposited in external repositories.

The source data of this paper are collected in the following database record: biostudies:S-SCDT-10_1038-S44319-024-00232-4.

## Peer review information

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

## Acknowledgements

The authors would like to thank Dr. Stefan Strack for providing Drp1 splice variant plasmids. We would like to thank Drs. Yisang Yoon, Janine Santos, and Patrick Kimes for helpful scientific discussions. We thank Royden Clark, Emmy Dier, Sara Shimko, and Ben Yankasky for their technical assistance. This work was supported by the U.S. Department of Defense CDMRP pilot award W81XWH-16-1-0117, TEAL award W81XWH-22-10252 and NIH R01CA242021 (to NH), NIH R01CA230628 (to KM and NH), NIH training grant 2T32HL110849-11A1 (to SW), NIH R35HL150778 (to MT), NIH S10OD023402 (to SLG) and NIH instrumentation grants S10RR025488, S10RR016236, and S10RR019003 (to SCW). Lauren Borho and Dr. Francesmary Mudugno kindly assisted as honest brokers to access patient specimens. The ProMark tissue bank is supported by NIH SPORE P50CA272218. This project used the Hillman Cancer Center Cytometry Facility, Animal Facility, In vivo Imaging Facility and

Cell and Tissue Imaging Facility that are supported in part by award P30CA047904.

## Author contributions

**Zaineb Javed**: Conceptualization; Data curation; Formal analysis; Validation; Investigation; Visualization; Methodology; Writing—original draft; Writing—review and editing. **Dong Hui Shin**: Data curation; Formal analysis; Investigation; Methodology. **Weihua Pan**: Data curation; Formal analysis. **Sierra R White**: Data curation; Formal analysis. **Amal Taher Elhaw**: Data curation; Formal analysis. **Yeon Soo Kim**: Data curation; Formal analysis. **Shriya Kamlapurkar**: Data curation; Formal analysis. **Ya-Yun Cheng**: Data curation; Formal analysis. **J Cory Benson**: Data curation; Formal analysis. **Ahmed Emam Abdelnaby**: Data curation; Formal analysis. **Rébécca Phaëton**: Conceptualization; Writing—review and editing. **Hong-Gang Wang**: Conceptualization; Writing—review and editing. **Shengyu Yang**: Conceptualization; Writing—review and editing. **Mara L G Sullivan**: Data curation; Methodology. **Claudette M St.Croix**: Resources; Methodology. **Simon C Watkins**: Resources; Methodology. **Steven J Mullett**: Data curation; Methodology. **Stacy L Gelhaus**: Resources; Methodology; Writing—review and editing. **Nam Lee**: Conceptualization; Writing—review and editing. **Lan G Coffman**: Conceptualization; Writing—review and editing. **Katherine M Aird**: Conceptualization; Writing—review and editing. **Mohamed Trebak**: Conceptualization; Funding acquisition; Writing—review and editing. **Karthikeyan Mythreye**: Conceptualization; Funding acquisition; Methodology; Writing—review and editing. **Vonn Walter**: Data curation; Formal analysis; Funding acquisition; Validation; Methodology; Writing—original draft; Writing—review and editing. **Nadine Hempel**: Conceptualization; Resources; Formal analysis; Supervision; Funding acquisition; Investigation; Visualization; Methodology; Writing—original draft; Project administration; Writing—review and editing.

Source data underlying figure panels in this paper may have individual authorship assigned. Where available, figure panel/source data authorship is listed in the following database record: biostudies:S-SCDT-10_1038-S44319-024-00232-4.

## Disclosure and competing interests statement

The authors declare no competing interests.

# Expanded View Figures

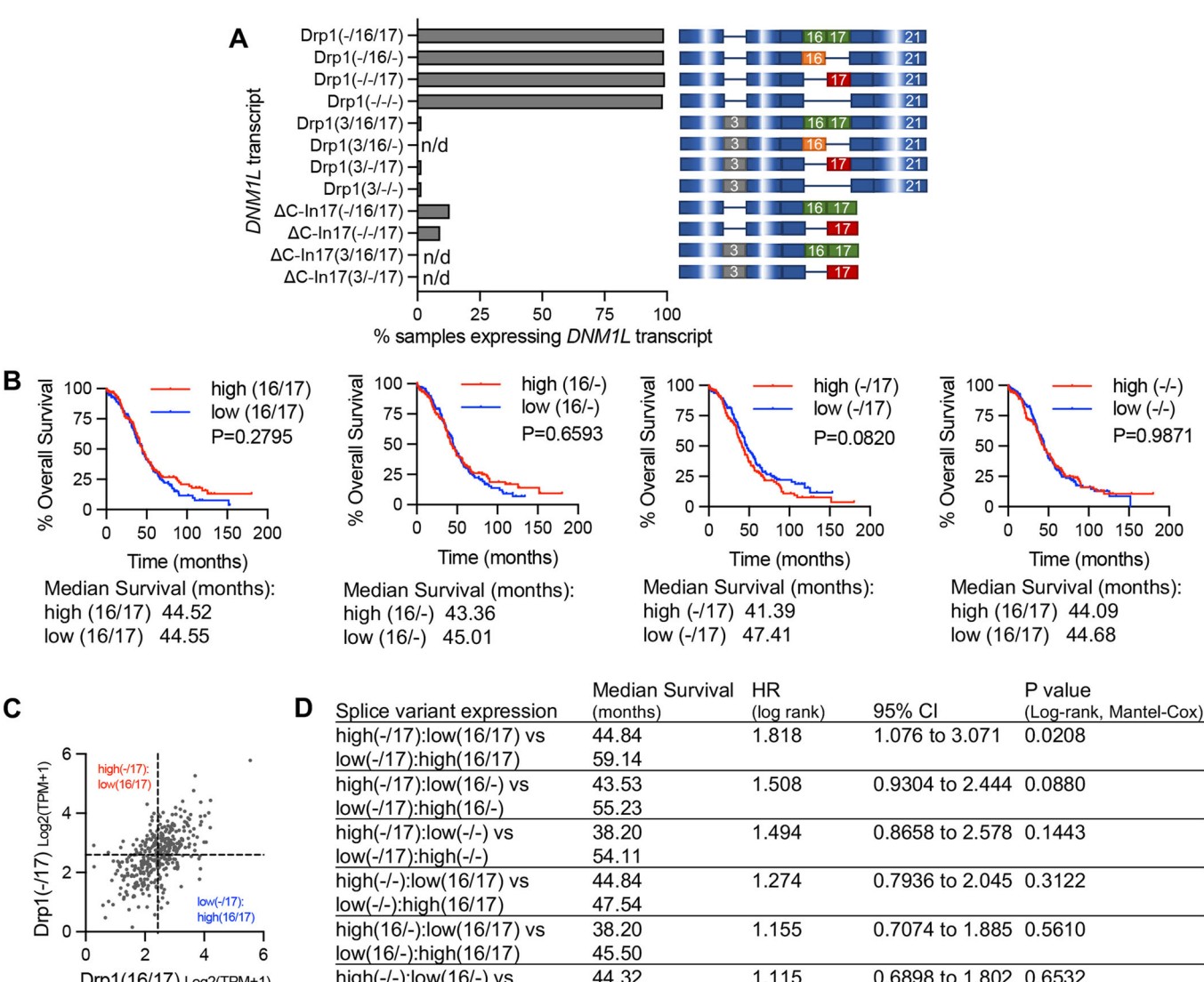

**Figure EV1. Drp1/*DNM1L* transcript variant expression in ovarian cancer specimens from TCGA.**

(**A**) Frequency of Drp1/*DNM1L* transcript variant expression, including alternatively spliced exons 3, 16, and 17 (3/16/17) transcripts and C-terminal truncation terminating in Intron 17 (ΔC-In17). Dash denotes exon is spliced out. Data represent the percentage of specimens displaying log2 TPM + 1 values >0.5 for each DNM1L variant. (**B**) Overall survival of TCGA patients based on DNM1L variant expression. Samples were split at median log2 TPM into high (*n* = 184) and low expression (*n* = 184; log-rank Mantel-Cox test). (**C**) Drp1(-/17) expression relative to Drp1(16/17; log2 TPM + 1). Mutually exclusive high and low expression of variant pairs is based on median log2 TPM + 1 expression cut-offs indicated by a dotted line. (**D**) Overall survival data of TCGA ovarian cancer patients grouped into mutually exclusive high/low expression of Drp1 transcript variant pairs. Low and high cutoffs are based on median expression. Patients with high Drp1(-/17) and low Drp1(16/17) expression display significantly decreased overall survival compared to patients with high Drp1(16/17) and low Drp1(-/17) transcript levels in their tumors.

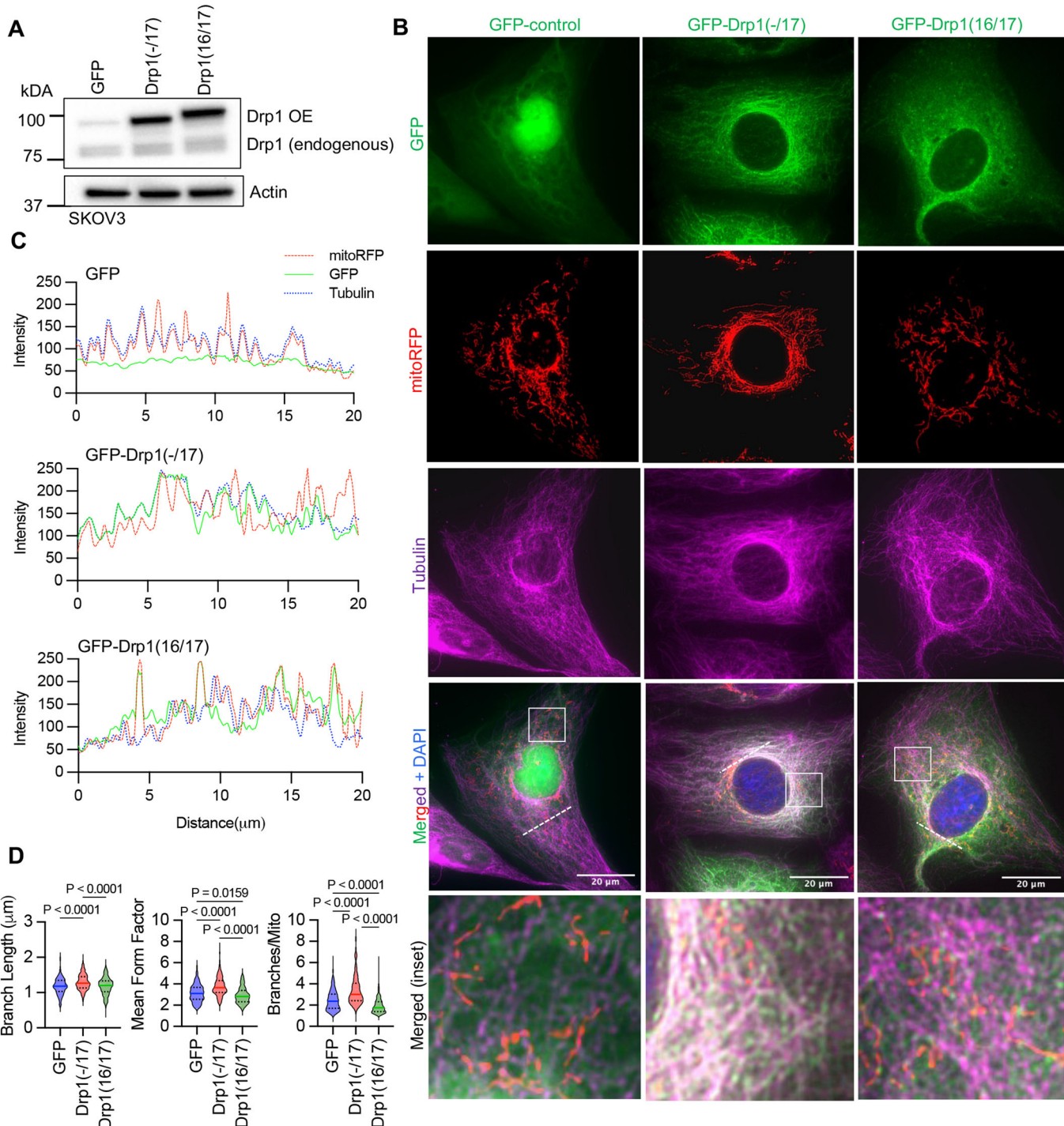

**Figure EV2.  Drp1(-/17) displays decreased association with mitochondria and increased localization to microtubules in SKOV3 cells.**

(A) Western blot analysis of Drp1 expression following transfection of GFP vector control, GFP-tagged Drp1(-/17) or Drp1(16/17) (overexpression: OE) in SKOV3 cells. (B) Representative epifluorescence images of mitochondrial morphology and Drp1 distribution in SKOV3 cells. (Green: GFP or GFP-tagged Drp1, Red: mito-RFP to label mitochondria, Magenta: anti-Tubulin antibody, Blue: DAPI). Drp1(-/17) expression is strongly co-localized with Tubulin, while Drp1(16/17) localizes to mitochondrial fission puncta. Scale bar: 20 μm. (C) Representative histograms of fluorescence intensity (dotted line in panel B images) illustrate that Drp1(-/17) (green) is more closely aligned with Tubulin (blue) and less so with mitochondria (red) in SKOV3 cells. In contrast, GFP-Drp1(16/17) fluorescence peaks coincide with mitochondrial (red) peaks, reflective of association with mitochondrial fission puncta. (D) Drp1(-/17) expressing SKOV3 cells display elongated and branched mitochondrial networks compared to cells expressing Drp1(16/17). Quantification of mitochondrial morphology was carried out using a mitochondria analyzer in ImageJ. (GFP control $n = 180$ cells, Drp1(-/17) $n = 224$ cells, Drp1(16/17) $n = 252$ cells, median + IQR, one-way ANOVA mean form factor $P < 0.0001$; branch length $P < 0.0001$; branches/mito $P < 0.0001$. Tukey's post test comparison $P$ values are shown).

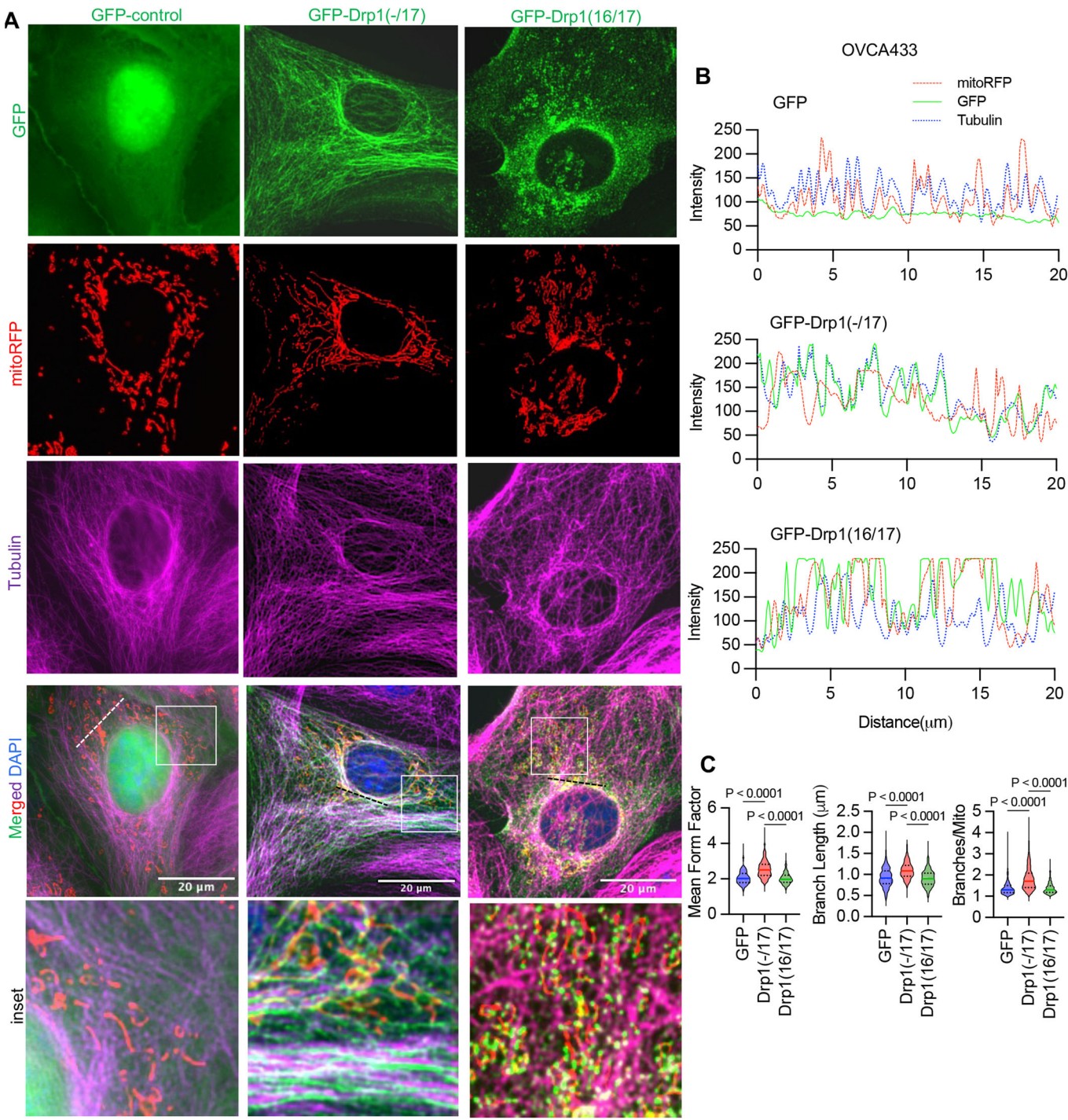

**Figure EV3. Drp1(-/17) displays decreased association with mitochondria in response to FCCP.**

(A) Drp1(-/17) preserves its localization with Tubulin upon treatment with the fission stimulus FCCP. In contrast, Drp1(16/17) associates with fission puncta at mitochondria in response to FCCP. Representative epifluorescence images are shown of mitochondrial morphology and Drp1 distribution after 30 min with FCCP treatment (1 μM) in OVCA433 cells. (Green: GFP or GFP-tagged Drp1, Red: mitochondria-targeted RFP, Magenta: anti-Tubulin, Blue: DAPI; Scale bar: 20 μm). (B) Representative histogram of fluorescence intensity of GFP-Drp1 (green) in conjunction with mitochondria (red) and Tubulin (blue), illustrates that GFP-Drp1(16/17) strongly overlaps with mitochondria following FCCP treatment (1 μM, 30 min). Conversely, Drp1(-/17) continues to show overlapping localization with tubulin rather than mitochondria. (C) OVCA433 cells expressing Drp1(16/17) or GFP control display decreased mitochondrial length and increased fragmentation compared to cells expressing Drp1(-/17) in response to FCCP. Quantification of mitochondrial morphological represented by three independent descriptors as analyzed by mitochondria analyzer in ImageJ. $n = 301$ cells from GFP control, $n = 285$ cells from Drp1(-/17), and $n = 287$ from Drp1(16/17) were analyzed (median + IQR, one-way ANOVA mean form factor $P < 0.0001$; branch length $P < 0.0001$ and branches/mito $P < 0.0001$. Tukey's post test comparison $P$ values shown).

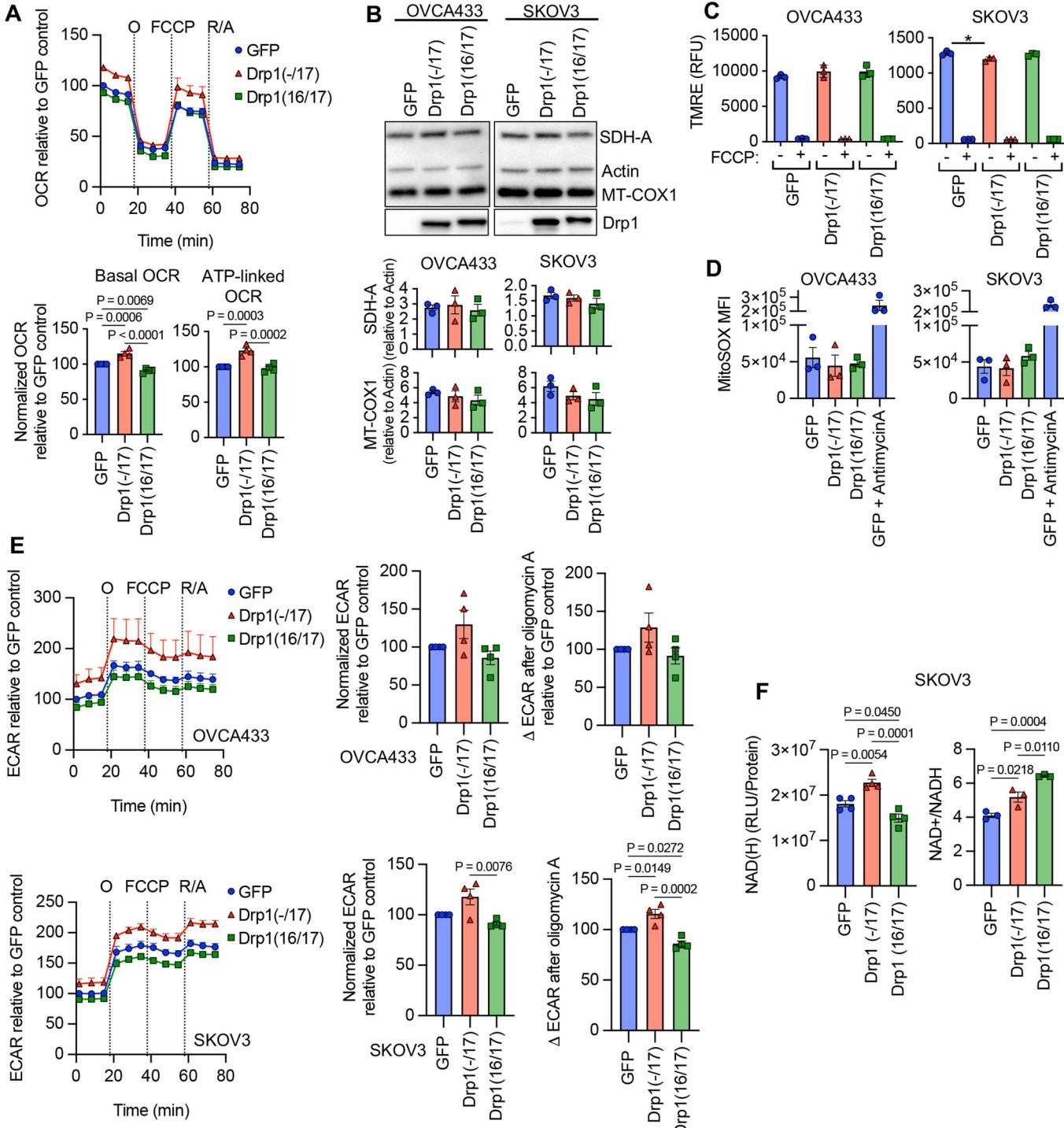

◀ **Figure EV4. Expression of Drp1(-/17) increases mitochondrial respiration and ECAR but does not affect levels of ETC components, mitochondrial membrane potential, or MitoSOX oxidation.**

(**A**) Expression of Drp1(-/17) increases oxygen consumption rates (OCR) in SKOV3 cells as assessed by Seahorse extracellular flux analysis and mitochondrial stress test (O: oligomycin A, R/A: rotenone/antimycin A; OCR is normalized to cell viability and expressed relative to GFP control). Basal OCR and ATP-linked OCR are increased in SKOV3 cells expressing Drp1(-/17) compared to Drp1(16/17). Data are expressed relative to GFP control (mean ± SEM of four biological replicates each derived from the average of 2–4 technical repeats, one-way ANOVA Basal OCR $P < 0.0001$; ATP-linked OCR $P = 0.0001$; Tukey's post test comparison $P$ values shown). (**B**) Levels of nuclear-DNA encoded SDH-A (Complex II) and mitochondrial DNA encoded COX1 (Complex IV) proteins are unchanged in both Drp1(-/17) and Drp1(16/17) expressing cells compared to GFP control cells. Data from one experimental replicate western blot is shown. Quantification of SDH-A and MT-COX1 protein expression normalized to β-Actin in OVCA433 and SKOV3 cells by densitometry using ImageJ. (mean ± SEM from lysates of three independent cultures, one-way ANOVA SDH-A expression; OVCA433 $p = 0.8407$, SKOV3 $P = 0.3893$, MT-COX1 expression; OVCA333 $P = 0.4876$, SKOV3 $P = 0.2842$). (**C**) Mitochondrial membrane potential was measured using TMRE (100 nM) at baseline and with FCCP treatment (10 μM, 30 min) in OVCA433 and SKOV3 cells expressing GFP control, Drp1(-/17) or Drp1(16/17) (mean ± SEM from three biological replicates each derived from the average of six technical repeats, one-way ANOVA of untreated cells OVCA433 $P = 0.3908$, SKOV3 $P = 0.0256$, Tukey's post test *$P = 0.0264$; one-way ANOVA comparison of FCCP treated cells OVCA433 $P = 0.3449$, SKOV3 $P = 0.1715$). (**D**) Drp1(-/17) and Drp1(16/17) overexpression in OVCA433 and SKOV3 cells did not alter the mean fluorescence intensity (MFI) of MitoSOX, a mitochondrial targeted dye susceptible to superoxide-mediated oxidation (mean ± SEM of MFIs from three biological replicates, one-way ANOVA OVCA433 $P = 0.7971$, SKOV3 $P = 0.3830$). Antimycin A (50 μM) was used as positive control. (**E**) ECAR traces derived from mitochondrial stress test (Figs. 4A OVCA433 and EV4A SKOV3). Basal ECAR and ΔECAR following oligomycin A inhibition of ATP-synthase were quantified and expressed relative to GFP control (mean ± SEM of 4 biological replicates each derived from the average of 2–4 technical repeats, one-way ANOVA, OVCA433 Basal ECAR $P = 0.0743$; OVCA433 ΔECAR $P = 0.1501$, SKOV3 Basal ECAR $P = 0.0088$; ΔECAR $P = 0.0003$; Tukey's post test comparison $P$ values shown). (**F**) Total NAD(H) levels are increased in response to Drp1(-/17) expression relative to SKOV3 cells expressing GFP control or Drp1(16/17), while the ratio of NAD+/NADH is significantly decreased (NAD(H): mean ± SEM of four biological replicates each derived from the average of three technical repeats, one-way ANOVA $P = 0.0002$; NAD+/NADH: mean ± SEM of four biological replicates each derived from the average of three technical repeats one-way ANOVA $P = 0.0004$; Tukey's post test $P$ values shown).

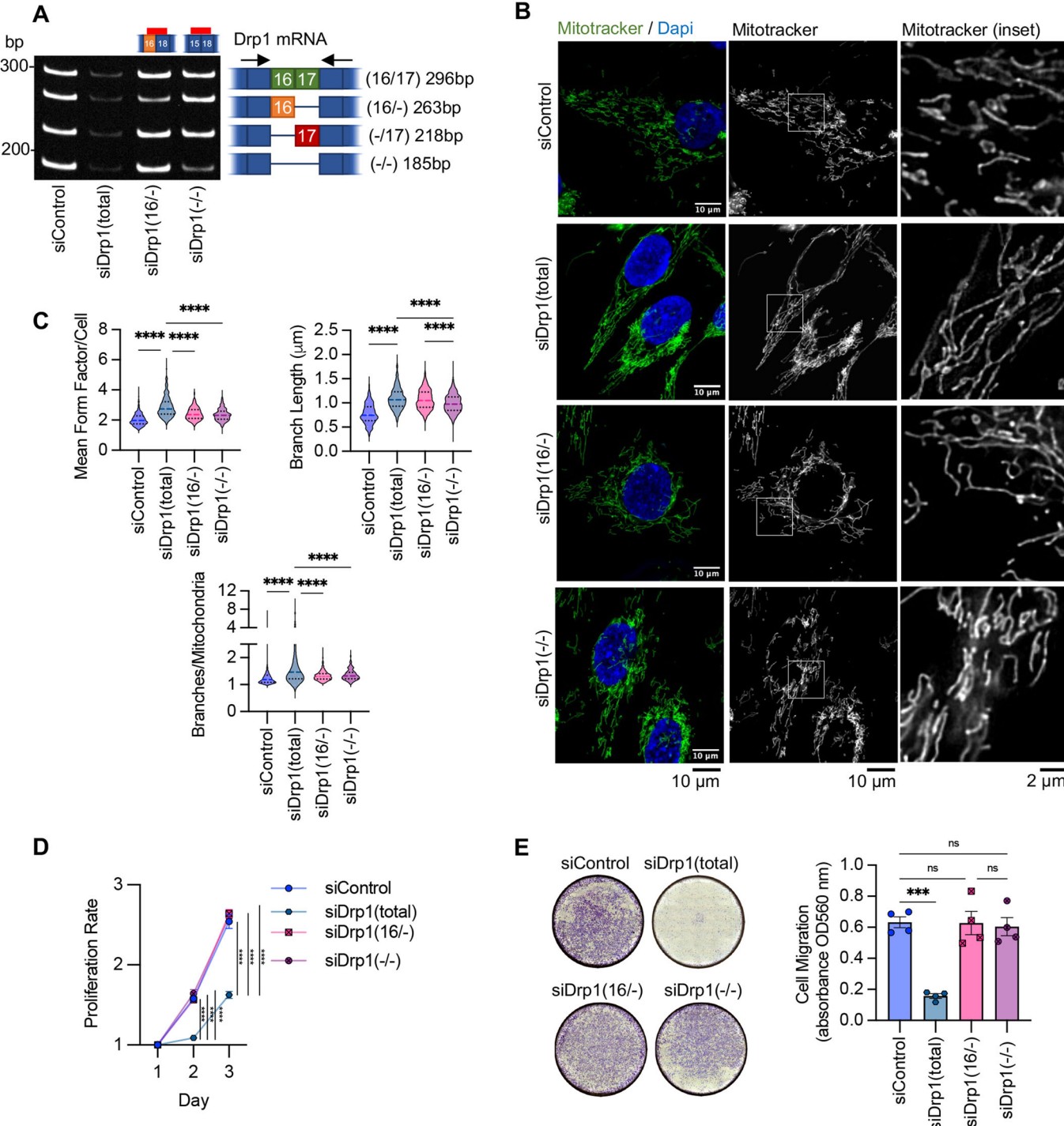

**Figure EV5. Specific knock-down of endogenous Drp1(−/−) and Drp1(16/-) variants and effects on mitochondrial morphology, cell proliferation, and migration.**

(A) Individual Drp1(−/−) and Drp1(16/-) variant-specific knockdown was achieved by use of siRNAs targeting Exon/Exon junction of splice variants (represented by red bars) and knock-down assessed by RT-PCR with primers flanking the variable domain region. (B) Representative epifluorescence images of mitochondrial morphology upon splice variant-specific siRNA Drp1 knockdown in SKOV3 cells (Green: mitotracker green, Blue: DAPI). Scale bar: 10 μm, inset 2 μm. (C) Quantification of mitochondrial morphological represented by three independent descriptors as analyzed by mitochondria analyzer in ImageJ. ($n = 560$ cells siControl, $n = 334$ cells siDrp1(total), $n = 630$ siDrp1(16/17), $n = 655$ siDrp1(-/17), $n = 555$ siDrp1(−/−)&(16/-); median + IQR, one-way ANOVA mean form factor $P < 0.0001$; branch length $P < 0.0001$ and branches/mito $P < 0.0001$. Tukey's post test was performed to assess differences between groups and analysis comparing groups to siDrp1(total) are shown, ****$P < 0.0001$). (D) Individual Drp1(−/−) and Drp1(16/-) variant-specific knockdown did not alter cell proliferation in SKOV3 cells, as there was no difference in proliferation rate compared to siControl cells. Cell proliferation was assessed by FluoReporter dsDNA quantification and proliferation rate expressed as an increase in the cell density relative to day 1 (mean ± SEM of four biological replicates each derived from the average of four technical repeats, two-way ANOVA group factor variance $P < 0.0001$, Tukey's post test ****$P < 0.0001$). (E) Cell migration was unchanged upon knock-down of either Drp1(−/−) or Drp1(16/-) splice variant in SKOV3 cells. Post Drp1 knock-down, cell migration was assessed using the Boyden chamber transwell assay and quantified by measuring the absorbance of the crystal violet staining of migrated cells. Images are representative of four independent assays ($n = 4$, mean ± SEM, one-way ANOVA $P < 0.0001$, Tukey's post test ***$P = 0.001$, with select comparisons shown).

