## [Peer Review File · EMBO Reports]

Drp1 splice variants regulate ovarian cancer mitochondrial dynamics and tumor progression

Zaineb Javed, Dong Hui Shin, Weihua Pan, Sierra White, Yeon Soo Kim, Amal Elhaw, Shriya Kamlapurkar, Ya-Yun Cheng, J Benson, Ahmed Abdelnaby, Rebecca Phaeton, Hong-Gang Wang, Shengyu Yang, Mara Sullivan, Claudette St. Croix, Simon Watkins, Steven Mullett, Stacy Gelhaus, Nam Lee, Lan Coffman, Katherine Aird, Mohamed Trebak, Karthikeyan Mythreya, Vonn Walter, and Nadine Hempel

Corresponding author(s): Nadine Hempel (nah158@pitt.edu)

Review Timeline:

Submission Date:	26th Feb 24
Editorial Decision:	28th Mar 24
Revision Received:	23rd May 24
Editorial Decision:	1st Jul 24
Revision Received:	13th Jul 24
Accepted:	26th Jul 24

Editor: Deniz Senyilmaz Tiebe

Transaction Report:

Dear Dr. Hempel,

Thank you for the submission of your research manuscript to our journal, which was now seen by two referees, whose reports are copied below.

Referees express interest in the proposed role of the presented Drp1 splice variants in ovarian cancer progression. However, they also raise significant concerns that need to be addressed to consider publication here.

Given these positive recommendations, we would like to invite you to submit a revised manuscript. Please revise your manuscript with the understanding that the referee concerns (as in their reports) must be fully addressed and their suggestions taken on board. Please address all referee concerns in a complete point-by-point response. Acceptance of the manuscript will depend on a positive outcome of a second round of review. It is EMBO reports policy to allow a single round of major experimental revision only and acceptance or rejection of the manuscript will therefore depend on the completeness of your responses included in the next, final version of the manuscript.

We realize that it is difficult to revise to a specific deadline. In the interest of protecting the conceptual advance provided by the work, we recommend a revision within 3 months. Please discuss the revision progress ahead of this time with me if you require more time to complete the revisions, or if you have questions or comments regarding the revision (also by video chat).

1. A data availability section providing access to data deposited in public databases is missing (where applicable).
2. Your manuscript contains statistics and error bars based on $n=2$. Please use scatter plots in these cases.

You can submit the revision either as a Scientific Report or as a Research Article. For Scientific Reports, the revised manuscript can contain up to 5 main figures and 5 Expanded View figures, and it should not exceed 27000 characters. If the revision leads to a manuscript with more than 5 main figures it will be published as a Research Article. In this case the Results and Discussion section should be separate. If a Scientific Report is submitted, these sections have to be combined. This will help to shorten the manuscript text by eliminating some redundancy that is inevitable when discussing the same experiments twice. In either case, all materials and methods should be included in the main manuscript file.

<<https://www.embopress.org/page/journal/14693178/authorguide#expandedview>>

4) a .docx formatted letter INCLUDING the reviewers' reports and your detailed point-by-point responses to their comments. As part of the EMBO publication's Transparent Editorial Process, EMBO reports publishes online a Review Process File (RPF) to accompany accepted manuscripts. This File will be published in conjunction with your paper and will include the referee reports, your point-by-point response and all pertinent correspondence relating to the manuscript.

<https://www.embopress.org/page/journal/14693178/authorguide#transparentprocess>

You are able to opt out of this by letting the editorial office know (emboreports@embo.org). If you do opt out, the Review Process File link will point to the following statement: "No Review Process File is available with this article, as the authors have

chosen not to make the review process public in this case."

5) a complete author checklist, which you can download from our author guidelines <https://www.embopress.org/page/journal/14693178/authorguide>. Please insert information in the checklist that is also reflected in the manuscript. The completed author checklist will also be part of the RPF.

6) Please note that all corresponding authors are required to supply an ORCID ID for their name upon submission of a revised manuscript (<<https://orcid.org/>>). Please find instructions on how to link your ORCID ID to your account in our manuscript tracking system in our Author guidelines <<https://www.embopress.org/page/journal/14693178/authorguide#authorshipguidelines>>

Additional information on source data and instruction on how to label the files are available: <https://www.embopress.org/page/journal/14693178/authorguide#sourcedata>

9) Our journal encourages inclusion of *data citations in the reference list* to directly cite datasets that were re-used and obtained from public databases. Data citations in the article text are distinct from normal bibliographical citations and should directly link to the database records from which the data can be accessed. In the main text, data citations are formatted as follows: "Data ref: Smith et al, 2001" or "Data ref: NCBI Sequence Read Archive PRJNA342805, 2017". In the Reference list, data citations must be labeled with "[DATASET]". A data reference must provide the database name, accession number/identifiers and a resolvable link to the landing page from which the data can be accessed at the end of the reference. Further instructions are available at <http://www.embopress.org/page/journal/14693178/authorguide#referencesformat>

12) Please also note our reference format: <http://www.embopress.org/page/journal/14693178/authorguide#referencesformat>

I look forward to seeing a revised version of your manuscript when it is ready. Please let me know if you have questions or comments regarding the revision.

Kind regards,

Deniz Senyilmaz Tiebe

Deniz Senyilmaz Tiebe, PhD
Scientific Editor
EMBO Reports

Referee #1:

In this study, Javed and colleagues investigate Drp1 splice variants the regulation of ovarian cancer tumorigenicity. Describing different Drp1 splice variants (as others have reported) they then demonstrate a correlation between Drp1 splice variant expression lacking exon 16 and poor outcome in ovarian cancer. Investigating this splice variant in detail, they describe impact on mitochondrial dynamics promoting various pro-tumorigenic effects (proliferation, survival, migration) - this is evidenced both through overexpression and RNAi mediated knockdown. Overall, the study is interesting, rigorous and the data largely support the authors' conclusions. I have a few minor points that should be addressed during revision (in no order of importance).

- Figure 1A, there is an apparent Drp1 splice variant (or possibly non-specific band) migrating well below 50kDa, can the authors comment on this ?
- Discussion of figure 2E, remove the word trended, it gives the erroneous impression of significant difference
- With respect to the overexpression studies, is Drp1 lacking delta 16 expected to act in a dominant negative manner on endogenous Drp 1 ? (phenotypes would suggest this is the case), this should be discussed.
- Figure 5D, can the survival data on these mice (with median survival time for each cell line) be plotted ? - would help visualize the survival difference.
- Figure 8, this data nicely complements the overexpression data, however it is lacking analysis of knockdown of specific isoforms on cisplatin/paclitaxel induced death (as in figure 6), this should also be included in these experiments to complete the data set.
- The nomenclature of the splice variants is confusing throughout, may be easier to refer to Drp1 lacking exon 16 as Drp1 (delta 16) for instance.

Referee #2:

In this manuscript the authors have studied the expression and effects of Drp1 splice variants in ovarian cancer cells. The authors detected a high expression of a Drp1 alternative splice variant lacking exon 16 in ovarian cancer patients showing a poor outcome. In addition, lack of exon 16 causes a reduced association of Drp1 to mitochondrial fission sites, and is associated with a more metastatic phenotype compared to full-length Drp1. The studies are of interest and are novel. The major weakness detected in the current version of this manuscript is the lack of information of how relevant is the lack of exon 16 at the level of the protein. Certainly this information should be provided to understand the relevance of the Drp1 splice variant.

1. Drp1 in ovarian cancer cells OVCA420 or OVCA433 is detected as a close doublet around 75 kDa, and under these conditions 4 different Drp1 variants were detected (Figure 1). It is key that the authors document the electrophoretic mobility of all 4 variants in PAGE-SDS gels.
2. In figure 2 the authors document a high expression of the variant lacking exon 16 in Fallopian tube or omental tumors. Are these effects statistically significant? It would be informative to show the data also as absolute values. In addition, it would be relevant to know whether the protein expression of Drp1 variant lacking exon 16 is also enhanced in ovarian tumors. What is the percentage of this variant compared to the total Drp1 protein?

3. Cells expressing Drp1 variant lacking exon 16 displaced most of the omental adipocytes. Was that displacement or that was due to a reduced proliferation-differentiation of adipocytes?
4. Figure 4A. The authors should indicate the statistics of amino acid concentrations. It is unclear whether any of the differences were statistically significant.
5. Images in Figure 7C too small to detect morphological changes in mitochondria. Show larger amplification of the images.
6. Authors used SKOV3 cells to repress Drp1 variant lacking exon 16. Surprisingly, this repression did not alter the expression of the different Drp1 bands. Based on the lack of differences in Drp1 expression, it is unclear how these cells show changes in mitochondrial morphology or in mitochondrial respiration.

We thank the editor and reviewers for their expertise and time in critically appraising our work and appreciate that they overall agreed on the novelty of our work. Below, we provide a point-by-point response to the reviewer's specific comments.

Referee #1:

In this study, Javed and colleagues investigate Drp1 splice variants the regulation of ovarian cancer tumorigenicity. Describing different Drp1 splice variants (as others have reported) they then demonstrate a correlation between Drp1 splice variant expression lacking exon 16 and poor outcome in ovarian cancer. Investigating this splice variant in detail, they describe impact on mitochondrial dynamics promoting various pro-tumorigenic effects (proliferation, survival, migration) - this is evidenced both through overexpression and RNAi mediated knockdown. Overall, the study is interesting, rigorous and the data largely support the authors' conclusions. I have a few minor points that should be addressed during revision (in no order of importance).

We thank reviewer 1 for their positive comments and their time and expertise in reviewing our work.

1. Figure 1A, there is an apparent Drp1 splice variant (or possibly non-specific band) migrating well below 50kDa, can the authors comment on this ?

RESPONSE: We apologize for omitting the labeling of this band as non-specific (n/s). This has now been corrected in Figure 1A & B:

Drp1-specific bands were first validated using an siRNA pool that targets all Drp1 splice variants. As shown in Appendix Figure S1G below, the polyclonal antibody (EMD Millipore ABT155) used in Figure 1A detects a number of non-specific bands. Non-specific bands detected by antibody Abcam 184247 are shown in Figure 1B above (please also see response to reviewer 2 comment 1).

Appendix FigureS1G: Protein variants identified in the range of 65-80kDa by western blotting are verified to be Drp1 using siRNA mediated knock-down in OVCA420 cells (Dharmacon siRNA pool). Non-specific bands (n/s) are not affected by siRNA targeting Drp1.

2. Discussion of figure 2E, remove the word trended, it gives the erroneous impression of significant difference

RESPONSE: Thank you for pointing this out. The description of Figure 2E has been amended on page 7 (lines 186-188) and this statement removed.

3. With respect to the overexpression studies, is Drp1 lacking delta 16 expected to act in a dominant negative manner on endogenous Drp1? (phenotypes would suggest this is the case), this should be discussed.

RESPONSE: Thank you for this insightful comment. We have postulated that this variant may have potential dominant negative functions, which may be due to sequestering the other variants away from mitochondria to the microtubules. While we presently do not have evidence for this, studies are planned to determine the interaction between the different Drp1 splice variants at the protein level and to assess how their interactions affect dimer and oligomer formation, and subcellular localization. We have now included the following in the discussion on page 18 (lines 538-541) to address this point: "Although further evidence is needed, such as assessment of heterodimer formation of Drp1 splice variants, it is possible that high expression of Drp1(-/17) could act as a dominant negative isoform by sequesters other splice variants away from mitochondria to microtubules."

4. Figure 5D, can the survival data on these mice (with median survival time for each cell line) be plotted? - would help visualize the survival difference.

RESPONSE: As suggested by reviewer 1, we have included the survival data in Figure 5E.

5. Figure 8, this data nicely complements the overexpression data, however it is lacking analysis of knockdown of specific isoforms on cisplatin/paclitaxel induced death (as in figure 6), this should also be included in these experiments to complete the data set.

RESPONSE: Thank you for this thoughtful suggestion. We have carried out these requested experiments and as shown below the differences in chemosensitivity to cisplatin and paclitaxel can also be observed by splice variant specific siRNA mediated knock-down. siRNA mediated knock-down of Drp1(-/17) increases cell death in response to both compounds, while knock down of Drp1(16/17) leads to abrogated caspase 3/7 activity in response to cisplatin and paclitaxel. These data have been added to Figure 8H and are described on page 13 (lines 395-398).

Figure 8H: Cells with single variant knock-down of Drp1(16/17) compared Drp1 (-/17) variant have reduced apoptotic response to cisplatin (5 µM) or paclitaxel (1 nM) treatment after 24 hours, as assessed using Caspase-Glo 3/7 assay (n=3, one-way ANOVA cisplatin p=0.0040; paclitaxel p=0.0016. Tukey's post test *p<0.05, **p<0.01).

6. The nomenclature of the splice variants is confusing throughout, may be easier to refer to Drp1 lacking exon 16 as Drp1 (delta 16) for instance.

RESPONSE: We acknowledge this issue and appreciate the suggestion. We wrestled with how to best depict the alternate splicing and multiple combinations of exons 3, 16 and 17. Due to the added complexity of the C-terminal truncations, which we termed deltaC, and the multiple combination of exon 3/16/17 splicing, we decided to maintain our original nomenclature. However, to better depict the variants we have now color coded these, provide more schematic representations throughout the figures (e.g. Fig 1E, 2A, 7E, 8A) and added descriptions of our used nomenclature throughout the text (e.g. Page 6, line 160-162).

Referee #2:

In this manuscript the authors have studied the expression and effects of Drp1 splice variants in ovarian cancer cells. The authors detected a high expression of a Drp1 alternative splice variant lacking exon 16 in ovarian cancer patients showing a poor outcome. In addition, lack of exon 16 causes a reduced association of Drp1 to mitochondrial fission sites, and is associated with a more metastatic phenotype compared to full-length Drp1. The studies are of interest and are novel. The major weakness detected in the current version of this manuscript is the lack of information of how relevant is the lack of exon 16 at the level of the protein. Certainly this information should be provided to understand the relevance of the Drp1 splice variant.

RESPONSE: We thank reviewer 2 for critically appraising our work and their valuable feedback. To address the valid concern related to the relevance of Drp1 splice variants at the protein level, please see our response to comment 1 below.

1. Drp1 in ovarian cancer cells OVCA420 or OVCA433 is detected as a close doublet around 75 kDa, and under these conditions 4 different Drp1 variants were detected (Figure 1). It is key that the authors document the electrophoretic mobility of all 4 variants in PAGE-SDS gels.

RESPONSE: To address this important point, we provide the following data on Drp1 splice variant expression at the protein level, as well as further evidence validating that the protein bands at MW 75kDa are indeed Drp1:

- Due to the close molecular weight of the 4 Drp1 protein variants, it is difficult to fully resolve these on SDS PAGE. However, we consistently detected multiple Drp1 protein bands around MW 75kDa in ascites-derived tumor cells from ovarian cancer patients (Fig. 1A) and in cancer cell lines OVCA420, OVCA433 and SKOV3 (Fig 1B, 7B, Appendix Fig. S1E, Appendix Fig. S7). These have now been more clearly labeled and non-specific bands pointed out (see response to reviewer 1 comment 1).
- Two independent Drp1 antibodies were used to detect these Drp1 protein variants (Fi. 1B, 7B, Appendix Fig. S7: Abcam 184247; Figure 1A, Appendix Fig. S1E: EMD Millipore ABT155), and independent siRNAs targeting the common regions shared by all transcript variants were used to validate that these proteins are indeed Drp1 (Fig. 1B, Appendix Fig. S1E: siRNApool; Fig. 7B, Appendix Fig. S7: in house-designed siRNA targeting exon 15).
- Importantly, using splice variant specific siRNAs we demonstrate selective knock-down of Drp1 splice variants at the protein level (Figure 7A, Appendix Fig. S7). In our revised manuscript we provide clearer representatives of these SDS-PAGE gels, as shown below.

Detection of DNM1L/Drp1 protein variants as validated by specific siRNA mediated knock-down targeting individual splice variants. n/s indicates non-specific bands detected by Drp1 antibody (Abcam 184247).

As can be seen on the right, OVCA433 cells show predominant expression of protein variants Drp1(-/17) and Drp1(16/17), which reflects the mRNA expression patterns of Drp1 splice variants in most ovarian cancer cell lines and specimens (Fig. 1F, 2C, 2D). Please note that the Drp1(16/-) and Drp1(16/17) variants differ in predicted molecular weight by only 1.3kDa (as Exon 17 contains only 11 amino acids). Thus, it was difficult to clearly resolve Drp1(16/-) from Drp1(16/17) on SDS-PAGE. However, their presence can be detected by splice variant specific siRNA targeting in SKOV3 cells (see above, orange and green arrows, respectively). Please note, although we found that the majority of ovarian cancer cells tend to have exon 3 spliced out, in some cell lines and tissues it is possible that the inclusion of Exon 3 could also contribute additional protein variants in this region.

2. In figure 2 the authors document a high expression of the variant lacking exon 16 in Fallopian tube or omental tumors. Are these effects statistically significant? It would be informative to show the data also as absolute values.

In addition, it would be relevant to know whether the protein expression of Drp1 variant lacking exon 16 is also enhanced in ovarian tumors. What is the percentage of this variant compared to the total Drp1 protein?

RESPONSE: As suggested by the reviewer, we have added the statistical comparison between normal fallopian tube tissues and matched ovarian or omental tumors in Figure 2D (shown below). The mRNA expression data of Drp1 splice variants (16/17), (16/-), (-/17) and (-/-) were derived from a single PCR with primers flanking the spliced exons 16 and 17, which allowed us to obtain accurate relative expression levels of these four splice variants in each specimen, even though the absolute Drp1 and actin levels were variable between patient samples. Please see Appendix Figure S2, which shows the gels for all PCRs from patient specimens represented in Figure 2, and the source data for Figure 2D which contains the absolute densitometry values. Please note that due to limited patient sample material, we were not able to extract protein and only able to obtain RNA from specimen shown in Figure 2.

Figure 2D: Representative RT-PCR (left) of DNMT1L variable domain splice variant expression from normal fallopian tube (N-F), and matched ovarian (T-OV) and omental tumors (T-OM). The relative expression of splice variant transcript Drp1(-/17) is consistently higher in ovarian tumor and omental tumor compared to matched normal fallopian tube specimens (blue lines indicate decreased expression, red lines indicate increased expression, and black lines indicate no change in expression relative to matched normal fallopian tube tissue, N-F vs T-OV n=13, N-F vs T-OM n=5; paired t-test).

3. Cells expressing Drp1 variant lacking exon 16 displaced most of the omental adipocytes. Was that displacement or that was due to a reduced proliferation-differentiation of adipocytes?

RESPONSE: We apologize for the erroneous use of the term displacement. Since we do not have direct evidence on tumor cell - adipocyte interactions, we have replaced this sentence with: "Omental tumors also differed morphologically, with cells expressing Drp1(-/17) forming a greater number of smaller and denser multi-lobular lesions along the omentum." (Pag 10 line297-298) However, this is an insightful thought that requires further investigation, given the observed changes in metabolism following Drp1 splice variant expression. Several studies have shown that adipocytes are necessary for homing and initial seeding of ovarian cancer cells to the omentum, by providing fatty acids as fuel

for cancer cell proliferation (e.g. Nieman et al Nat Medicine 2011). An active area of investigation is focused on assessing the changes that occur in adipocytes present in this tumor microenvironment.

4. Figure 4A. The authors should indicate the statistics of amino acid concentrations. It is unclear whether any of the differences were statistically significant.

RESPONSE: According the reviewer's suggestion, we have included the statistical comparisons in Appendix Figure S4:

Appendix Figure S4: Untargeted metabolite analysis of OVCA433 cells expressing GFP, Drp1(-/17) or Drp1(16/17) (Metabolite levels relative to internal standard (iSTD); n=4, one-way ANOVA; Tukey's post test *p<0.05, **p<0.01, ***p<0.001, ****p<0.0001)

5. Images in Figure 7C to small to detected morphological changes in mitochondria. Show larger amplification of the images.

RESPONSE: Thank you for this valuable suggestion. Higher magnification images have been added to Figure 7C and Extended View Figure 5B.

6. Authors used SKOV3 cells to repress Drp1 variant lacking exon 16. Surprisingly, this repression did not alter the expression of the different Drp1 bands. Based on the lack of differences in Drp1 expression, it is unclear how these cells show changes in mitochondrial morphology or in mitochondrial respiration.

RESPONSE: As can be seen in Figure 7A and 7B, an siRNA targeting only splice variant Drp1(-/17) was able to specifically decrease Drp1(-/17) expression at the RNA and protein levels. These data show that we are able to specifically decrease expression of individual splice variants using siRNAs designed to target individual splice variants, or use a combination of these siRNAs to enrich for endogenous expression of specific splice variants, as shown in Figure 8A(ii) & F. Please also see the blot above in response to reviewer 2, comment 1: lanes labeled with siDrp1(-/17), with the red arrows

demonstrating knock-down of Drp1(-/17). Using this siRNA approach were able to consistently achieve specific knock-down of Drp1 splice variants and this was validated for each experiment where the effects on mitochondrial morphology and function in response to splice variant specific knock-down were tested.

Other revisions:

- We noticed and corrected several spelling and grammar mistakes.
- Figure 5C originally showed the average of all technical replicates, which has been replaced to show the comparisons of the means of technical replicates from 3 independent experiments.
- In the original manuscript Figure 5D contained a representative image of 5 mice from the wrong cage (albeit it being from the same experimental group). This has now been corrected, and all images for the complete set of 10 animals per group can be accessed in the Source data.

Dear Nadine,

Thank you for submitting your revised manuscript. It has now been seen by both original referees.

I apologize for this unusual delay in getting back to you. It took longer than anticipated to receive the referee reports.

As you can see, the referees find that the study is significantly improved during revision and recommend publication. However, I need you to address the points below before I can accept the manuscript.

- Please provide 3-5 keywords for your study. These will be visible in the html version of the paper and on PubMed and will help increase the discoverability of your work.
- Please rename the Conflict of Interests section as "Disclosure Statement and Competing Interests".
- Please remove the Author Contributions section from the manuscript.
- We noted the following regarding the figure callouts - missing callouts: Fig. 7E, Table 3; Fig. 10 called out but does not exist.
- Please add a Table of Contents and page numbers to the Appendix file.
- Tables 1-3 need to be placed between main and EV figures.
- Our production/data editors have asked you to clarify several points in the figure legends:
 - o Please note that the exact p values are not provided in the legends of figures 3d, f; 4b, d; 5a-c, e; 6a-b; 7d; 8b-h; EV 2d; EV 3c; EV 4a; EV 5c-e.
 - o Please note that in figures 4b, d; 5g; 8e-f; EV 3c; EV 4a, f; there is a mismatch between the annotated p values in the figure legend and the annotated p values in the figure file that should be corrected.
 - o Although 'n' is provided, please describe the nature of entity for 'n' in the legends of figures 1f; 2b; 4a-b, d; 5a; 6b; 8b-d, h; EV 4a-f; EV 5d.

Thank you again for giving us to consider your manuscript for EMBO Reports, I look forward to your minor revision.

Kind regards,

Deniz

--

Deniz Senyilmaz Tiebe, PhD
Editor
EMBO Reports

Referee #1:

The authors have comprehensively addressed all my comments

Referee #2:

The authors have adequately revised the manuscript based on my prior comments, and I do think that is suitable for publication.

Dear Dr. Senyilmaz Tiebe,

We thank you and the reviewers for their time and effort in reviewing our revised manuscript. We appreciate that the reviewers found that our work was significantly improved during revision and recommend publication. Below we provide a point by point response to your additional comments:

- Please provide 3-5 keywords for your study. These will be visible in the html version of the paper and on PubMed and will help increase the discoverability of your work.

The following Keywords have been added to the manuscript:

Drp1, *DNM1L*, mitochondrial fission, ovarian cancer, alternative splice variants

- Please rename the Conflict of Interests section as "Disclosure Statement and Competing Interests".

This has been renamed.

- Please remove the Author Contributions section from the manuscript.

This has been removed.

- We noted the following regarding the figure callouts - missing callouts: Fig. 7E, Table 3; Fig. 10 called out but does not exist.

Fig 7E and table 3 are now mentioned in the text. The erroneous mention of Fig 10 has been removed.

- Please add a Table of Contents and page numbers to the Appendix file.

A table of contents has been added.

- Tables 1-3 need to be placed between main and EV figures.

The tables have been removed from the text and uploaded as separate files.

- Our production/data editors have asked you to clarify several points in the figure legends:

o Please note that the exact p values are not provided in the legends of figures 3d, f; 4b, d; 5a-c, e; 6a-b; 7d; 8b-h; EV 2d; EV 3c; EV 4a; EV 5c-e.

The exact p-values have been directly added to the figure panels for figures 3d, 3f, 4b, 4d, 5b, 5c, 5e, 6b, 6d, 7d, 8b, 8c, 8d, 8g, 8h, EV2d, EV3c, EV4a, EV4e, EV4f,

The exact p-values have been added to the figure legends for 5a, 6a, 8e, 8f, EV5c, EV5d, EV5e.

o Please note that in figures 4b, d; 5g; 8e-f; EV 3c; EV 4a, f; there is a mismatch between the annotated p values in the figure legend and the annotated p values in the figure file that should be corrected.

These have been corrected by adding exact p-values, as stated above.

o Although 'n' is provided, please describe the nature of entity for 'n' in the legends of figures 1f; 2b; 4a-b, d; 5a; 6b; 8b-d, h; EV 4a-f; EV 5d.

These descriptions have been added to the figure legends.

1f: mean +/- SEM from 3 independent cultures.

2b: log₂(TPM+1) values from 368 individual Ovarian Serous Cystadenocarcinoma TCGA patient samples

4a: mean +/- SEM of 4 biological replicates each derived from the average of 2-4 technical repeats

4b: mean +/- SEM of 4 biological replicates each derived from the average of 2-4 technical repeats

5a: mean +/- SEM of 3 biological replicates each derived from the average of 4 technical repeats

6b: mean +/- SEM of 3 biological replicates each derived from the average of 3-4 technical repeats

8b-d: mean +/- SEM of 4 biological replicates each derived from the average of 3 technical repeats, unpaired t-test P values shown

8h: mean +/- SEM of 3 biological replicates each derived from the average of 3 technical repeats

EV4a: mean +/- SEM of 4 biological replicates each derived from the average of 3 technical repeats

EV4b: mean +/- SEM from lysates of 3 independent cultures

EV4c: mean +/- SEM from 3 biological replicates each derived from the average of 6 technical repeats

EV4d: mean +/- SEM of MFIs from 3 biological replicates

EV4e: mean +/- SEM of 4 biological replicates each derived from the average of 2-4 technical repeats

EV4 f: NAD(H): mean +/- SEM of 4 biological replicates each derived from the average of 3 technical repeats; NAD⁺/NADH: mean +/- SEM of 4 biological replicates each derived from the average of 3 technical repeats

EV5d: mean +/- SEM of 4 biological replicates each derived from the average of 4 technical repeats

Dr. Nadine Hempel
University of Pittsburgh
UPMC Hillman Cancer Center
5051 Centre Ave
Pennsylvania 15213
United States

Dear Dr. Hempel,

Thank you for submitting your revised manuscript. I have now looked at everything and all is fine. Therefore, I am very pleased to accept your manuscript for publication in EMBO Reports.

Congratulations on a nice work!

Kind regards,

Deniz Senyilmaz Tiebe

--

Deniz Senyilmaz Tiebe, PhD
Editor
EMBO Reports

--
